# Self-healing polyurethane-elastomer with mechanical tunability for multiple biomedical applications in vivo

Chenyu Jiang [1,5], Luzhi Zhang[2,5], Qi Yang [1], Shixing Huang [1], Hongpeng Shi[1], Qiang Long[1], Bei Qian[1], Zenghe Liu[2], Qingbao Guan [2], Mingjian Liu[3], Renhao Yang[4], Qiang Zhao[1✉], Zhengwei You [2✉] & Xiaofeng Ye[1✉]

The unique properties of self-healing materials hold great potential in the field of biomedical engineering. Although previous studies have focused on the design and synthesis of self-healing materials, their application in in vivo settings remains limited. Here, we design a series of biodegradable and biocompatible self-healing elastomers (SHEs) with tunable mechanical properties, and apply them to various disease models in vivo, in order to test their reparative potential in multiple tissues and at physiological conditions. We validate the effectiveness of SHEs as promising therapies for aortic aneurysm, nerve coaptation and bone immobilization in three animal models. The data presented here support the translation potential of SHEs in diverse settings, and pave the way for the development of self-healing materials in clinical contexts.

[1] Department of Cardiovascular Surgery, Ruijin Hospital, Shanghai Jiao Tong University, School of Medicine, Shanghai, China. [2] State Key Laboratory for Modification of Chemical Fibers and Polymer Materials, Shanghai Belt and Road Joint Laboratory of Advanced Fiber and Low-dimension Materials (Donghua University), College of Materials Science and Engineering, Donghua University, Shanghai, China. [3] Department Neuro Surgery, Ruijin Hospital, Shanghai Jiao Tong University, School of Medicine, Shanghai, China. [4] Department Orthopedics Surgery, Ruijin Hospital, Shanghai Jiao Tong University, School of Medicine, Shanghai, China. [5] These authors contributed equally: Chenyu Jiang, Luzhi Zhang. ✉email: zq11607@rjh.com.cn; zyou@dhu.edu.cn; xiaofengye@hotmail.com

nspired by living tissues, research on self-healing materials has advanced over several years, and such materials have shown great potential in biomedicine[1,2]. However, most self-healing processes require external stimuli (such as heat and UV light), which are harmful to living bodies[3,4]. Recently, autonomous self-healing materials have attracted increasing attention because their healing conditions are consistent with the conditions of use[5]. However, the use of toxic catalysts or metal ions in self-healing materials is also unacceptable for bioapplications[6]. Although a few self-healing hydrogels have been investigated for use in biomedical engineering, their limited biodegradability and mechanical properties hinder their applications[7]. Furthermore, in vivo applications utilising the self-healing properties of materials have not yet been investigated. Owing to their biomimetic mechanical properties in soft tissues and excellent resilience that is compliant with the dynamic in vivo environment, elastomers such as heart assistance devices and medical catheters have been widely used in biomedicine[8,9]. Therefore, biocompatible and biodegradable self-healing elastomers (SHEs) possess great potential for biomedical applications but have not been reported. Herein, we designed a new type of autonomously SHE composed of polyurethanes based on dynamic dimethylglyoxime–urethane groups. These SHEs showed mechanical tunability, biocompatibility and biodegradability and demonstrated high efficiency in treating certain clinical conditions, such as aneurysms, peripheral nerve amputation and bone immobilisation, because of their excellent self-healability under physiological conditions (Fig. 1).

Suture is the most basic technique in traditional surgery, and it is also an important means of tissue reconstruction and fixation of prostheses. However, suture, which need long-term training, is a time-consuming and repetitive work, reducing the efficiency of operations. Moreover, due to the rapid advancement of minimally invasive surgery, it is extremely difficult to suture in a narrow surgical field. The emergence of SHE could avoid the process of surgical suture, achieving real-time shaping of material during the surgery, greatly simplifying the surgical procedure especially under certain circumstances like endoscopic operation or intracranial operation, etc., improving the efficiency of operations and reducing the chance of surgical trauma to adjacent tissues. In this article, we will demonstrate the surgical process changes brought about by self-healing materials via using three surgery disease models including aneurysm, peripheral injury, and sternum immobilisation.

Aneurysm, peripheral nerve amputation and bone immobilisation, which have been challenging for doctors and patients for a long time, are common but not easily cured diseases in clinical scenarios. Aneurysms are considered as time bombs that are a risk to human life at all ages[10]. Due to the luminal dilatation of aneurysms, blood vessel haemodynamics transform from streamline to turbulent flow, which leads to endothelial cell dysfunction and aneurysm expansion[11–14]. Peripheral nerve coaptation after amputation is used to be achieved by suturing which has several drawbacks such as time-consuming operations, extra-epineural axonal growth and increased odds of fascicular

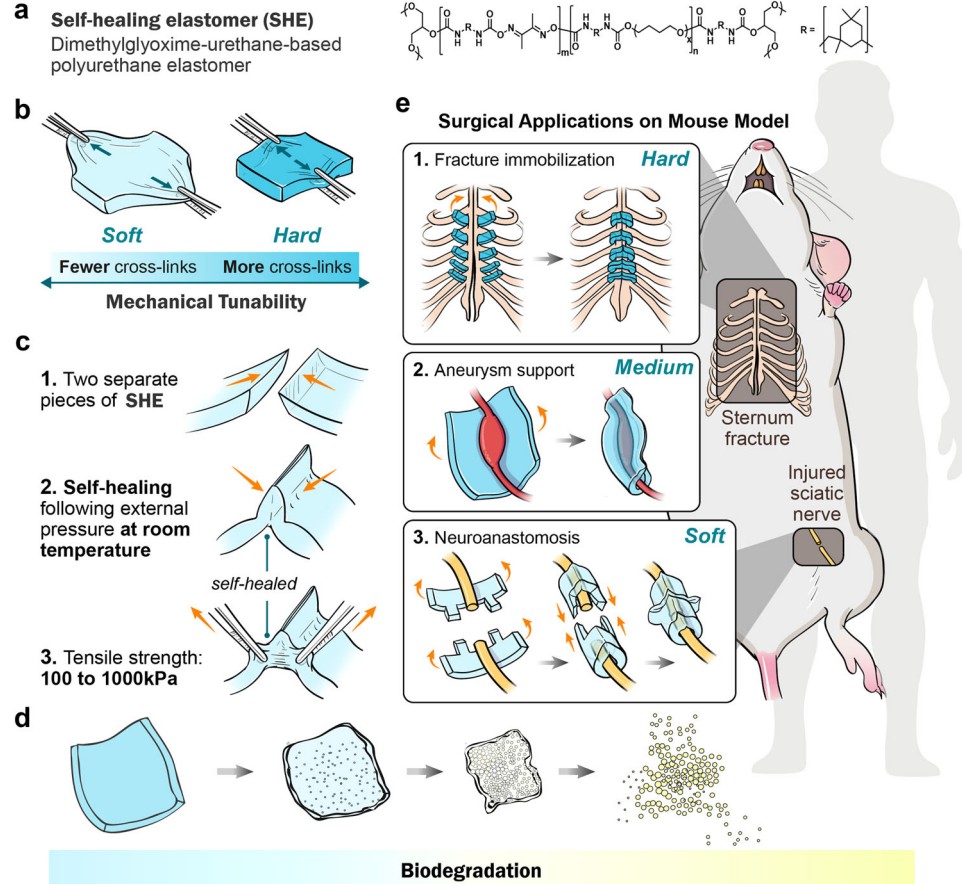

**Fig. 1 Pattern diagram of chemical structural formula, tunable mechanical and self-healing properties, and in vivo biomedical applications of SHEs. a** Abbreviation and full name of SHEs with its chemical structural formula. **b** Presentations of tunable mechanic by adjusting the degree of crosslink. **c** Pattern diagram of self-healing property of SHEs. **d** Pattern diagram of biodegradability of SHEs. **e** Three applications of SHEs in biomedical area in vivo. (From up to down) Fractured sternum bone immobilisation, wrapping aneurysm for limiting the progress, and LEGO-playing peripheral nerve coaptation. SHE self-healing elastomer.

misconnection[15,16]. Sternum immobilisation after median sternotomy is a critical procedure after cardiac surgery[17]. Traditional stainless steel wire cerclage for immobilisation can cause the sternum dehiscence due to the cutting action of metal wires, and can hardly provide good sternal stability leading to serious complications such as pain, osteomyelitis and even respiratory complications[18,19]. In addition, residual undegradable steel wires may lead to future problems such as chronic bone non-union and deep sternal wound infection[20,21].

In this study, we design and fabricate a family of mechanically tunable, biocompatible and biodegradable SHEs for biomedical applications. Importantly, we explore and demonstrate in vivo applications of the self-healing ability of SHEs to address the poorly resolved clinical scenarios described above. We take advantage of the tunable mechanical and self-healing properties of SHEs to correct haemodynamics, improve endothelial function and limit the progress of aneurysms. Additionally, we provide a novel, convenient and efficient approach for nerve coaptation by utilising the self-healing property of SHEs, making coaptation as simple as playing with LEGO® bricks and much more efficient than suturing. Moreover, we replace metal wires with a biodegradable SHEs to achieve sternum immobilisation and avoid the shortcomings mentioned above.

## Result

**Hydrolysis behaviour of the oxime–urethane bond.** Here, we designed model compound **a** to study the hydrolysis behaviour of oxime–urethane groups. When compound **a** was mixed with water, the NMR spectrum showed the characteristic peaks of compound **a** and water except for the deuterated solvent (day 0) (Supplementary Fig. 1). After 60 days, the characteristic peaks of the hydrolysis products compound **b** and compound **c** appeared in the NMR spectrum (day 60). We defined the proton integration of the $=CH_2$ group in the 2-isocyanatoethyl methacrylate moiety ($\delta$ 5.68 and 6.06 ppm) and that of the NH group ($\delta$ 7.78 ppm), which corresponded to one molecule of compound **a**, as 4 protons and 2 protons, respectively. Because to the protons of the $=CH_2$ group were far from the reaction centre, the signals at $\delta$ values of 5.68 and 6.06 ppm remained unchanged during the reaction. On the basis of the decrease in the relative integration of the peak at $\delta$ 7.78 ppm, we calculated that approximately 21% of the oxime–urethane bonds of compound **a** were hydrolysed.

**Synthesis and characterisation of SHEs.** The molecular structure and schematic of SHEs are shown in Fig. 2a and b. The SHEs were synthesised from DMG, PTMEG, IPDI and glycerol, in the absence of a catalyst. The molecular structures of the SHEs were confirmed by ATR-FTIR spectroscopy. The peaks at 3318 and 1713 cm$^{-1}$ corresponded to N–H and C=O bonds in carbamate units, respectively, indicating the formation of urethane groups. There were negligible peaks at 2264 cm$^{-1}$, which corresponded to N=C=O groups, revealing that the monomer IPDI had reacted completely. According to the results of dynamic mechanical analysis (DMA), the glass transition temperatures of SHE0, SHE0.2, SHE0.5, SHE1 and SHE2 were −35.7, −21.2, −18.1, −7.2 and 20.6 °C, respectively (Supplementary Fig. 2B). The DSC curves of the SHEs did not show crystallisation or melting peaks from −60 to 100 °C (Supplementary Fig. 3A). As seen from the rheological test, the addition of the crosslinking points to the SHEs enhanced the mechanical strength of the polymer at both 25 and 37 °C (Supplementary Fig. 3A). Water contact angle measurements were used to determine the contact angles (92 −99°) of the SHEs, revealing the hydrophobicity of the materials (Supplementary Fig. 3B).

**Mechanics, biodegradability and biocompatibility analysis of the SHEs.** Uniaxial tensile tests showed that the tensile strength of the SHEs ranged from 33 kPa to 4.383 MPa, Young's modulus ranged from 172 kPa to 3.724 MPa, and the breaking elongation ranged from 506% to 3295% (Figs. 2d, e and S5A). The self-healing properties were tested after cutting strips of the SHEs into two pieces and rejoining again at room temperature for 5 min without any external stimuli. The healed SHEs showed sufficient mechanical properties for application in different in vivo scenarios (Figs. 2f, g and Supplementary Fig. 3B). The mechanical properties of the SHEs in the wet state were similar to those of the SHEs in the dry state (Supplementary Fig. 4). To demonstrate the elasticity of the SHEs, cyclic tensile tests at a strain of 20% were performed (Fig. 2h). As the number of cycles increased, the tensile strength of the non-crosslinked SHE0 gradually decreased, and could not recover to its original state even after 30 min of relaxation. With an increasing degree of the crosslinking degree, the reduction in the mechanical strength gradually decreased, and the recovered strength gradually increased after 30 min of relaxation. All SHEs exhibited biodegradability in PBS containing CE with an activity of 300 U mL$^{-1}$ at 37 °C. A total of ~7% of the mass of all SHEs was lost within 9 days (Fig. 2i). Additionally, after subcutaneous implantation in C57BL/6 mice, the crosslinked SHEs (0.2–2) maintained their original shape on day 35 post implantation (Supplementary Fig. 6), and scanning electron microscopy showed micro-holes on the surfaces of the SHE sheets, which implied the degradation in vivo (Fig. 2l). The safety of the SHEs needs to be determined prior to in vivo applications. Liver and renal function were not altered on the day 35 after SHE implantation compared to that of the sham group (Supplementary Figs. 7 and 8). The proliferation of fibroblast cells cultured with the SHE2 was satisfactory, particularly compared to that of cells cultured with the elastomer containing the catalyst as a negative control and PCL as a positive control (Fig. 2j and k). In addition, the inflammatory response of SHEs (using SHE2 for inflammation analysis) was minimal in the both acute and chronic subcutaneous inflammation analyses, while the elastomer containing the catalyst showed a severe acute inflammatory response at day 7 post of implantation (Supplementary Fig. 9).

**Aneurysm limitation.** To explore the application value of the mechanical support of SHEs, we tentatively used three types of crosslinked SHEs (SHE2, SHE1, and SHE0.5) to limit the progress of aortic aneurysms. However, at 14 days post-surgery, MRI scanning showed no flow void effect in the lumen of the aorta in the SHE2 group, while a flow void effect was observed in the lumen of the aorta the other four groups (Fig. 3B and Supplementary Fig. 12). Furthermore, SHE2 was observed to result in injury of the aorta in the presence of collateral circulation under general observation (Supplementary Fig. 12). Therefore, the pathology analysis of the SHE2 could not proceed further. Moreover, to the naked eye, the shape of the aorta in the aneurysm group was deformed significantly towards the shape of a spinning spindle, while the aortas of animals in the SHE1, SHE0.5 and non-self-healing groups maintained a tubular shape despite being slightly thicker than the aortas of the sham group (Fig. 3b and Supplementary Figs. 9 and 18). Furthermore, the eSHE1 and eSHE0.5 groups showed satisfactory limitation effects on 3-day established aneurysms, as seen by comparing the MRI images obtained before and after the wrapping (Supplementary Figs. 15 and 16). Statistically, from the diameter of the aorta calculated from MRI data those of the aneurysm group were significantly higher than those parameters for the other four groups (Fig. 3c and d). Additionally, the diameter of non-self-healing group was higher than that of SHE0.5 group with statistical difference

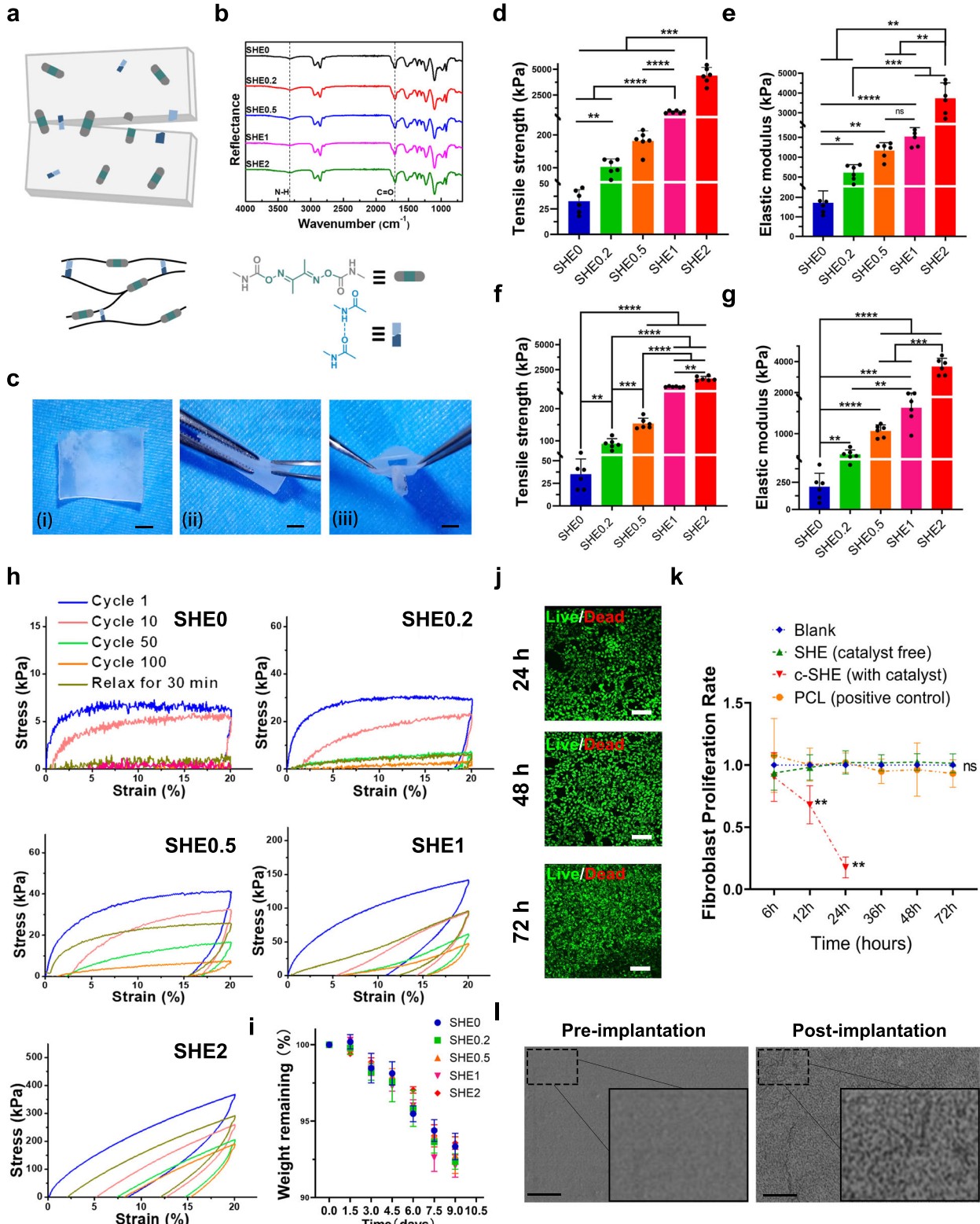

(Supplementary Fig. 18D). Moreover, there was no significant difference in the diameter of before and after wrapping in the eSHE1 and eSHE0.5 groups (Supplementary Fig. 15C). Destruction of the elastic layer (containing elastin[22]) is the main force exacerbating the progress of aneurysm expansion. EVG staining in this experiment showed a relatively intact structure in both the SHE1 and SHE0.5 groups, although the structure was poorer than

that in the sham group (Fig. 3e). Based on the quantity of layers and integrity of elastin, 4 elastin grades were presented (Fig. 3f and Supplementary Fig. 18E). The statistical histogram of the elastin grades showed no significant difference among the sham, SHE0.5 groups and non-self-healing group, and no significant difference between the aneurysm and SHE1 groups, even though the grade of the SHE1 group was lower than that of the aneurysm group.

**Fig. 2 Pattern diagram of structure of SHEs, mechanical test, ex vivo degradation and biocompatibility analysis. a** Molecular structure, schematics and **b** ATR-FTIR spectra of SHEs. **c** In vitro self-healing property demonstration, scale bar = 1 mm. Histogram shows the **d** tensile strength and **e** elastic modulus of SHEs, $n = 6$ in each group. Histogram shows the **f** tensile strength and **g** elastic modulus of healed SHEs (self-healing for 5 min at room-temperature), $n = 6$ in each group. **h** Cyclic tensile curves of SHEs at 20% strain. There was no waiting time between two consecutive cyclic tensile (cycle 1–cycle 100). The film was then allowed to relax for 30 min at 25 °C before the 101st cyclic tensile test. **i** In vitro enzymatic degradation of SHEs in CE solution at 37 °C, $n = 3$ in each group. **j** Live and dead staining of areolar fibroblast cell co-cultured with SHE (using SHE2) on 24, 48 and 72 h, each experiment was repeated three times. scale bar = 150 μm. **k** Proliferation rate of areolar fibroblast cell co-cultured with c-SHE (with catalyst, as a negative control), SHE (using SHE2) and PCL (as a positive control) respectively on 6, 12, 24, 48 and 72 h ($n = 5$ in each group). **l** Scanning electron micrograph of SHE (using SHE2) before subcutaneous degradation experiment (pre-implantation) and on day 35 of subcutaneous degradation experiment (post-implantation). The surface of SHE arose the appearance of micro-holes after 35-day implantation in vivo. Each experiment was repeated three times, scale bar = 2 μm. Data were presented as mean ± s.d. Brown–Forsythe ANOVA test with Dunnett's multiple comparison test **d**–**g** was used for comparing tensile strength and elastic modulus in groups before and after self-healing. Tensile strength before self-healing (**d**): SHE0, SHE0.2, SHE0.5 and SHE1 compared to SHE2, ***$p = 0.0003, 0.0003, 0.004$ and $0.0009$, respectively; SHE0 compared to SHE0.5, ***$p = 0.003$; SHE0 compared to SHE0.2, **$p = 0.0034$; SHE0.5 compared to SHE1, ****$p < 0.01$; Elastic modulus before self-healing (**e**): SHE0 and SHE0.2 compared to SHE2, **$p = 0.006$ and $0.008$, respectively; SHE0.5 and SHE1 compared to SHE2, **$p = 0.0019$ and $0.0044$, respective; SHE0 compared to SHE0.2, *$p = 0.0153$; SHE0 compared to SHE0.5 **$p = 0.002$; SHE0.5 compared to SHE1, ns $p = 0.1318$; SHE0 compared to SHE1, ****$p < 0.0001$; Tensile strength after self-healing (**f**): SHE0.5, SHE1 and SHE2 compared to SHE0, ****$p < 0.0001$; SHE0.5 compared to SHE0.2, ***$p = 0.0005$; SHE1 and SHE2 compared to SHE0.2, ****$p < 0.0001$; SHE1 and SHE2 compared to SHE0.5, ****$p < 0.0001$; SHE0 compared to SHE0.2, **$p = 0.0015$; Elastic modulus after self-healing (**g**): SHE0.5, SHE1 and SHE2 compared to SHE0, ****$p < 0.0001$; SHE0.5 and SHE1 compared to SHE2, ***$p = 0.0002$ and $0.0002$, respectively; SHE0 compared to SHE0.2, **$p = 0.048$; SHE0 compared to SHE0.5, ****$p < 0.0001$; SHE0 compared to SHE1, ***$p = 0.0002$; SHE0.2 compared to SHE1, **$p = 0.0045$. Ordinary one-way ANOVA test with Tukey's multiple comparisons test (**k**) was used for evaluation of fibroblast proliferation rate. SHE (catalyst free) group and PCL group compared to blank group, ns $p > 0.999$; SHE (catalyst free) group compared to PCL group, ns $p = 0.995$; Elastomer (with catalyst) compared to blank group, **$p = 0.002$; Elastomer (with catalyst) compared to SHE (catalyst free) group, **$p = 0.002$; Elastomer (with catalyst) compared to PCL group, **$p = 0.0024$; Source data are provided as a Source Data file. SHE self-healing elastomer, PCL polycaprolactone, c-SHE containing catalyst self-healing elastomer.

However, the elastin grade was not different among the aneurysm, eSHE1 and eSHE0.5 groups (Supplementary Fig. 17A and D). Endothelial cells participate in many physiological activities to maintain the normal structure and function of the vessel wall[23]. Dysfunctional endothelial cells present a decreased eNOS expression. The immunohistochemical staining of eNOS was stronger in the sham, SHE1, SHE0.5 and non-self-healing groups than in the aneurysm group and the foldchange of eNOS transcription level was lower in the aneurysm group than other four groups (Fig. 3g and Supplementary Fig. 14A). Additionally, the statistical plot showed that the expression of eNOS in the SHE1 and SHE0.5 groups was statistically higher than the expression of eNOS in the aneurysm group, while there was no significant difference between the SHE1 and SHE0.5 groups, even though the expression of eNOS was higher in the SHE0.5 group (Fig. 3h). The expression of eNOS in the eSHE1 and eSHE0.5 groups was significantly higher than aneurysm group (Fig. S17B and E). Collagens, including type I and type III collagens, maintain the integrity and stability of the artery wall[24], while arterial injury may disrupt the balance between the two types of collagens[25]. From the immunofluorescence (IF) staining of collagens in the aortic wall, we found that the ratio of collagen I/III in the aneurysm group was decreased significantly. In addition, the ratio decreased, although not significantly, in the SHE1 group compared to the aneurysm group, while the ratio of the SHE0.5 group was significantly higher than that of the aneurysm group and the non-self-healing material group was higher than that of the aneurysm group without significant difference (Fig. 3i–l and Supplementary Fig. 18G). Additionally, the mean collagen I/III ratio in the eSHE1 and eSHE0.5 was higher than the mean ratio of collagen in the aneurysm group, but the difference was not significant (Supplementary Fig. 17F). The level of inflammation was lower in group of SHE1, SHE0.5 and non-self-healing than aneurysm group (Supplementary Fig. 13). The neovascularization and apoptosis of vascular smooth muscle cell were also lower in group of SHE1, SHE0.5 and non-self-healing than aneurysm group. But compared to SHE0.5 group, the neovascularization and apoptosis of vascular smooth muscle cell were increased in non-self-healing group (Supplementary Fig. 13).

**Nerve coaptation**. The design of nerve coaptation in this part of the study is summarised in a schema (Fig. 4a). To achieve nerve coaptation completed by SHE0.2, the pair of SHE0.2 must be trimmed into a certain shape similar to a castle wall and healed together by the feet of the trimmed SHE0.2 s. The paper-cut models shown in Fig. 4b simulated the process of SHE0.2 coaptation. Compared to the time required for suture coaptation, that required for nerve coaptation using SHE0.2 was greatly shortened by 3/4, achieving the goal of rapid coaptation (Fig. 4c and Supplementary Fig. 21). In general, after sciatic nerve injury, the five ipsilateral toes cannot be consciously abducted. Hence, the degree of toe splay can indirectly reveal the degree of nerve recovery. After 6 weeks of coaptation, the degree of ipsilateral toe splay in the SHE0.2 group was more obvious than in the suture group and was better than that in the defect group (Fig. 4d and Supplementary Fig. 22A). Additionally, we can evaluate the functional recovery of the sciatic nerve by calculating the sciatic-nerve function index (SFI) from gait analysis. At 6 weeks post coaptation, nerve recovery was better in the SHE0.2 group than in the suture, fibrin-glue and PCL-conduit groups, with significant differences among them according to the statistical histogram of the SFI (Fig. 4e and Supplementary Fig. 22B). The implanted SHEs still protected the nerve as a conduit on week 6 post coaptation (Supplementary Fig. 21C). HE staining of transverse sections of the distal coaptation nerve showed a high density of myelinated fibres with many capillaries in the suture, SHE0.2, fibrin glue and PCL conduit groups, while the quantity of capillaries was not significantly different among those groups (Fig. 4i, j and Supplementary Fig. 26A). In general, myelinated axons present strong immunostaining for the neurofilament protein NF200[26], while the calcium-binding protein-S100β present in Schwann cells around axons exhibits neurotrophic activity, thus enhancing neurite outgrowth and the survival of neurons in peripheral nerve injury[27]. In our study, the distal anastomotic nerve fibres were positive for NF200 and S100β under a fluorescence microscope in all groups (Fig. 4K). Furthermore, longitudinal axis sections at the coaptation showed continuous nerve fibres positive for NF200 and S100β under fluorescence microscopy (Supplementary Fig. 24A and B). The percent area of NF200-and S100β-positive

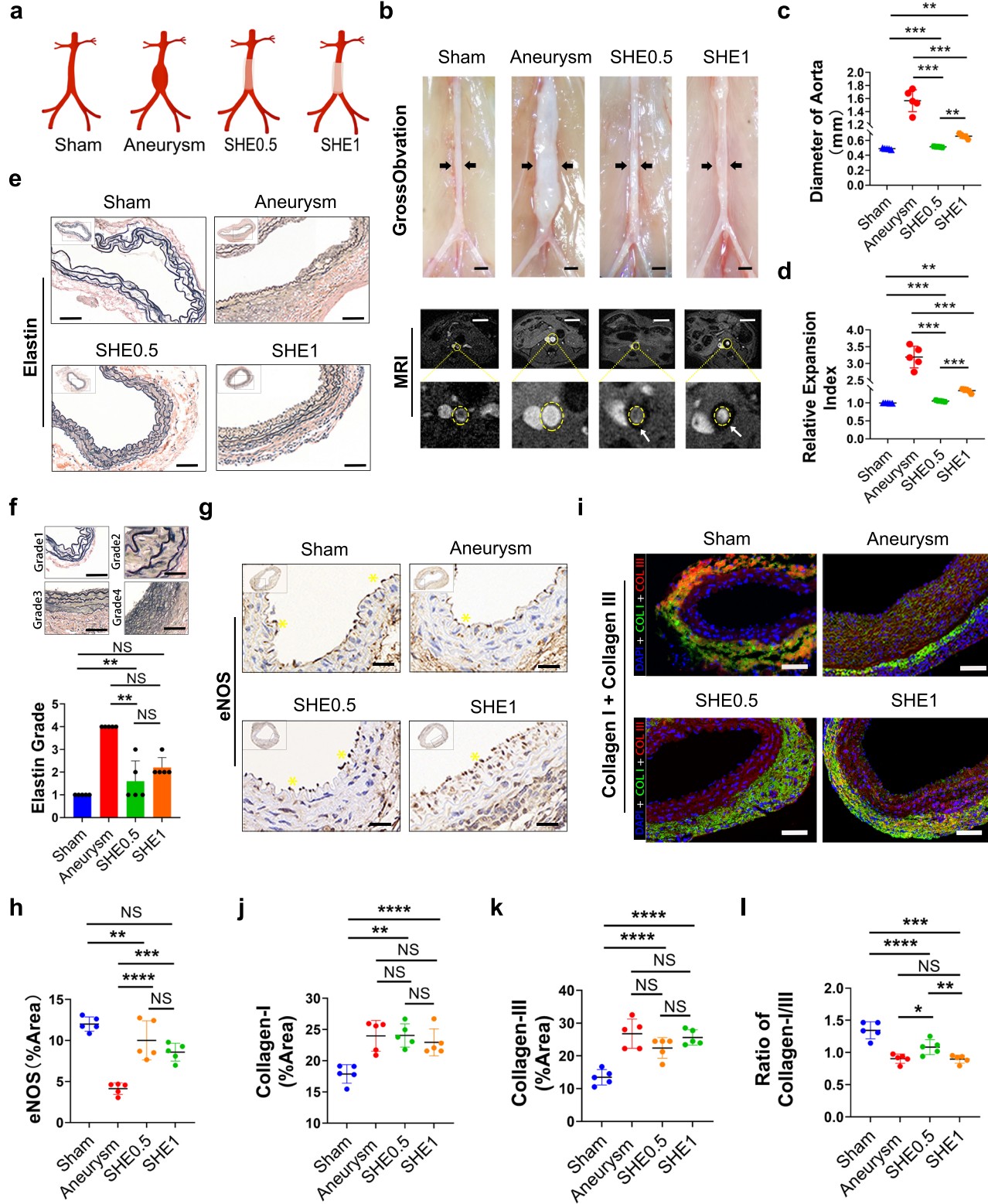

nerve fibres in the SHE0.2 group was significantly higher than that in the suture, fibrin and PCL-conduit groups, while there was no difference compared to that in the sham group (Fig. 4m, 4n, and Supplementary Fig. 26B, C). Histomorphometry revealed that the myelin thickness and number of axons counted under TEM were still higher in the SHE0.2 group than in the suture, fibrin-glue and PCL-conduit groups but lower in the SHE0.2 group than in the sham group (Supplementary Fig. 29). Dual

immunostaining labelling of S100β and caspase-3 revealed that the counting of apoptotic Schwann cell was significant lowest in SHE group than other groups (Supplementary Fig. 31A and B). Quantitative Real-time PCR data showed that the transcription of caspase-3 mRNA was also significant lowest in SHE group than other groups (Supplementary Fig. 31C). Retrograde labelling by fluorogold showed that the number of motor neurons in the spinal anterior horn was higher in the SHE0.2 group than in the

**Fig. 3 Pattern diagram of grouping, Gross observation and MRI angiography of abdominal aorta, HE staining, EVG staining of aorta wall structure and immunofluorescence staining of aorta wall on collagen I/III. a** Pattern diagram shows the grouping of this part. **b** Gross observations and MRI scanning results of aorta in each group. The blood vessels pointed by the black arrow in the gross observation pictures are the abdominal aortas, scale bar = 1 mm. Of the MRI scanning, inside the yellow dotted circle is the abdominal aorta, while the black circular area pointed by the white arrow is the elastomer wrapped around the artery, scale bar = 5 mm. **c** Statistical plot of the maximum diameter of abdominal aorta in each group ($n = 5$ in each group). **d** Statistical plot of the relative expansion index of abdominal aorta in each group ($n = 5$ in each group). **e** Elastica van gieson (EVG) staining on wall of aorta. Elastin inside the normal arterial wall is stained with dark brown multi-circle intact rings, each experiment was repeated three times. Elastin inside the aneurysm is broken and incomplete, scale bar = 50 µm. **f** Panel **f** shows the subjective criteria for elastin grade and statistical histogram of elastin grade, $n$ = 5 in each group, scale bar = 20 µm. **g** Endothelial nitric oxide synthase (eNOS) staining performed on aorta wall. The dark brown particles on the surface of vascular endometrium and pointed by the yellow * are nitric oxide synthase, scale bar = 30 µm. **h** Statistical plot of the expression of eNOS of abdominal aorta in each group ($n = 5$ in each group). **i** Immunofluorescence staining of collagen I (green) and collagen III (red) on aorta wall, scale bar = 50 µm. **j–l** Panels show the statistical plot of the expression of collagen I, collagen III and the ratio between collagen I and III, respectively ($n = 5$ in each group). Data were presented as mean ± s.d. Brown–Forsythe ANOVA test with Dunnett's multiple comparison test (**c**, **d**) was used for comparing diameter of aorta in groups. Diameter of aorta (**c**): Sham compared aneurysm, ***$p = 0.0006$; Sham compared to SHE0.5, **$p = 0.0046$; Sham compared to SHE1, ***$p = 0.0004$; Aneurysm compared to SHE0.5, ***$p = 0.0007$; Aneurysm compared to SHE1, **$p = 0.0013$; SHE0.5 compared to SHE1, **$p = 0.0022$; Relative expansion index (**d**): Sham compared aneurysm, ***$p = 0.0005$; Sham compared to SHE0.5, **$p = 0.0044$; Sham compared to SHE1, ***$p = 0.0007$; Aneurysm compared to SHE0.5, ***$p = 0.0005$; Aneurysm compared to SHE1, ***$p = 0.001$; SHE0.5 compared to SHE1, ***$p = 0.0005$. Kruskal–Wallis test with Dunn's multiple comparisons test (**f**) was used for elastin grading. Sham compared to aneurysm, ***$p = 0.0009$; Sham compared to SHE0.5, ns $p > 0.9999$; Sham compared to SHE1, ns $p = 0.2593$; Aneurysm compared to SHE0.5, *$p = 0.0251$; Aneurysm compared to SHE1, ns $p = 0.4614$; SHE0.5 compared to SHE1, ns $p > 0.9999$. Ordinary one-way ANOVA test with Tukey's multiple comparisons test (**h**, **j**, **k**, **l**) was used for evaluation of eNOS expression and collagen expression in groups. eNOS expression (**h**): Sham compared aneurysm, **$p = 0.0011$; Sham compared to SHE0.5, **$p = 0.001$; Sham compared to SHE1, ***$p = 0.0058$; Aneurysm compared to SHE0.5, ns $p > 0.9999$; Aneurysm compared to SHE1, ns $p = 0.8488$; SHE0.5 compared to SHE1, ns $p = 0.8233$; Collagen-I expression (**j**): Sham compared aneurysm, **$p = 0.0011$; Sham compared to SHE0.5, **$p = 0.001$; Sham compared to SHE1, ***$p = 0.0058$; Aneurysm compared to SHE0.5, ns $p > 0.9999$; Aneurysm compared to SHE1, ns $p = 0.8488$; SHE0.5 compared to SHE1, ns $p = 0.8233$; Collagen-III expression (**K**): Sham compared aneurysm, ****$p < 0.0001$; Sham compared to SHE0.5, **$p = 0.0021$; Sham compared to SHE1, ****$p < 0.0001$; Aneurysm compared to SHE0.5, ns $p = 0.1792$; Aneurysm compared to SHE1, ns $p = 0.9389$; SHE0.5 compared to SHE1, ns $p = 0.4169$; Ratio of collagen-I/III expression (**l**): Sham compared aneurysm, ****$p < 0.0001$; Sham compared to SHE0.5, **$p = 0.004$; Sham compared to SHE1, ****$p < 0.0001$; Aneurysm compared to SHE0.5, ns $p = 0.0525$; Aneurysm compared to SHE1, ns $p = 0.9995$; SHE0.5 compared to SHE1, *$p = 0.0425$. Source data are provided as a Source Data file. eNOS endothelial cell nitric oxide synthase, NS no significance.

suture, fibrin-glue and PCL-conduit groups, while there was no significant difference between the SHE and fibrin-glue groups (Supplementary Fig. 28B and D). In addition, the number of DRG neurons in the SHE0.2 group was higher than that in the suture, fibrin-glue and PCL-conduit groups, and there was no significant difference between the SHE and fibrin-glue groups (Supplementary Fig. 28A and C). Moreover, the SHEs did not induce a more chronic inflammatory response (Supplementary Fig. 25). Sciatic nerve injury in particular induces atrophy of the gastrocnemius muscle, one of the target muscles of sciatic nerves. Atrophy is usually demonstrated by a decreased fibre size and muscle weight. Masson staining of the transverse section of the gastrocnemius muscle showed a large quantity of collagen deposited around the thin muscle fibres in the defect groups. Additionally, almost no muscle fibrosis was observed in the sham, suture, SHE0.2, fibrin-glue and PCL-conduit groups (Fig. 4f and Supplementary Fig. 30). Statistically, the muscle fibre size was not different among the sham, SHE0.2, fibrin-glue and suture groups, while the muscle fibre size in these groups was significantly higher than that in the conduit and defect group (Fig. 4g and Supplementary Fig. 30). Consistently, the wet weight ratio of the gastrocnemius muscle in rats was not different among the suture, SHE0.2 fibrin-glue and PCL-conduit groups, however, the weights in these groups were significantly higher than the weights in the defect group (Fig. 4h).

**Sternum immobilisation.** The pattern diagram presents the concept of sternum immobilisation with a SHE2 (Fig. 5a). Figure 5B shows the surgical procedure of sternum immobilisation with SHE2 on rats. On week 6 post immobilisation, the rats underwent X-ray imaging and then, the sternums of the rats were harvested for further analysis. The X-ray showed a high-density shadow associated with the healing sternum in each group and

the gross observations clarified the X-ray results (Supplementary Fig. 32). The harvested tissues were sliced into transverse sections along the short axis of the sternum, and the sections were stained with HE and Masson trichrome after decalcification. HE staining showed relatively intact cortical bone in the steel wire group and the SHE2 group, with an irregular shape compared to the intact cortical bone in the sham group (Fig. 5c). Additionally, the cortical bone thickness was not significantly different among the groups (Fig. 5d). Increased collagen deposition plays an important role in the healing of bone fractures[28,29]. Masson staining showed large-scale deposition of collagen in the steel wire group and the SHE2 group (Fig. 5c). In addition, the statistical histogram of collagen deposition showed a significant difference between the sham and SHE2 groups (Fig. 5e). Considering the small size of rats, we also performed a test on the pigs. Based on the instant spiral X-ray imaging after surgery, the porcine sternum was immobilised by SHE2 and the fracture edges were also aligned (Fig. 5f).

## Discussion

A self-healing material is a kind of smart polymer that can achieve mutual healing between broken parts by reconstructing the molecular structure from the fractured form to the intact form. The mechanism of self-healing involves spontaneous or inductive reformulation of covalent or noncovalent bonds. In the last decade, self-healing materials have been explored for use in biomedical applications such as drug delivery, biosensors, and tissue engineering[30,31]. However, at present, the studies of truly utilising the properties of self-healing to treat certain clinical conditions have not been reported, likely due to their poor biocompatibility and limited self-healing efficiency of such materials in the physiological environments. Most current studies on the bio-application of self-healing materials have ignored their self-healing properties in vivo. In this study, we designed and synthesised a new family of SHEs with superior self-healability, and these materials were utilised as the key to addressing several clinical challenges in vivo.

SHEs are based on dynamic oxime–urethane bonds, which are a new type of dynamic covalent bond and have just recently been used to prepare dynamic polymers[5,32–35]. However, the use of toxic catalysts and metal ions[5,32,33] and the

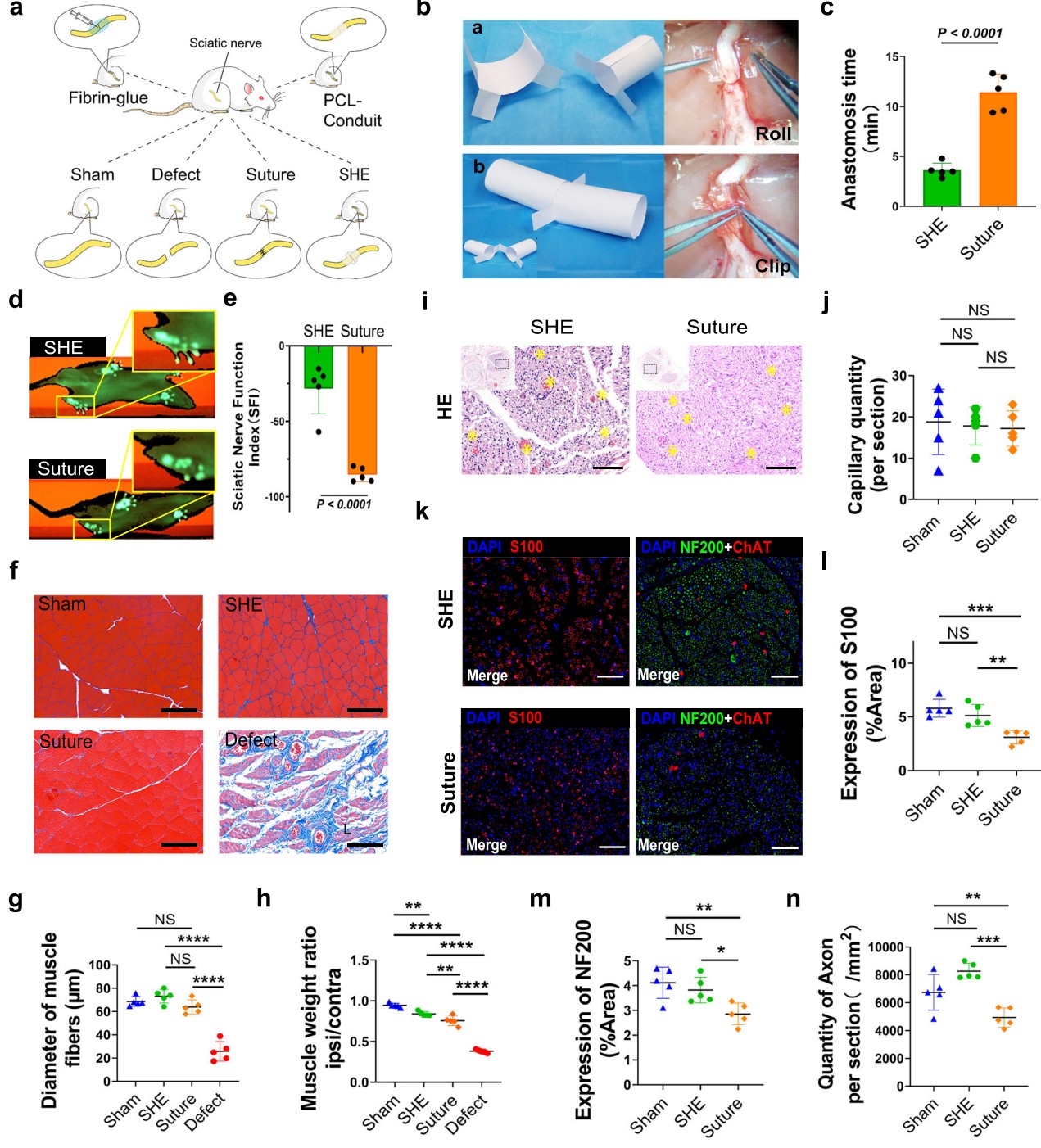

need for high self-healing temperatures[34,35], greatly limit the use of these materials in the body. SHEs were readily synthesised by a one-pot polycondensation from commercial reagents (DMG, PTMEG, IPDI and glycerol) in the absence of catalysts, which improved the safety of the SHEs for biomedical applications. The involved starting materials involved including PTMEG, IPDI and glycerol have been widely used in the synthesis of biomaterials[36]. We completed both in vitro and in vivo biocompatibility analyses prior to further in vivo applications. Favourable biocompatibility in vitro and in vivo verified the good biocompatibility of SHEs (Fig. 2j, k and Supplementary Fig. 9), indicating their safety for further bioapplications.

Degradability is also an important parameter for biomaterials. However, traditional polyurethanes are difficult to biodegrade. SHEs were designed as biodegradable polyurethanes, with the oxime–urethane bond playing a key role. First, we used small model molecules to study the hydrolysis behaviour of oxime–urethane

bonds. NMR spectroscopy was used to monitor the reaction (Supplementary Fig. 1). After 60 days, compounds **b** and **c** were formed as a result of the hydrolysis of the oxime–urethane bond. We calculated that ~21% of the oxime–urethane bonds were hydrolysed. Next, the biodegradability of the SHEs was also demonstrated by the results of in vitro enzymolysis and in vivo subcutaneous experiments (Fig. 2i and l).

Because of their highly dynamic oxime–urethane bonds and hydrogen bonds, the SHEs exhibited self-healing properties. In addition, the chain of PTMEG is flexible, and the high steric hindrance of the IPDI unit inhibits crystallisation. As a result, SHEs exhibited low $T_g$ values (for SHE0, SHE0.2, SHE0.5, and SHE1 below −0 °C) and did not crystallise, ensuring sufficient chain mobility at room temperature and facilitating reversible breakage and reformation of the oxime–urethane bonds and hydrogen bonds for self-healing. More importantly, a large number of dimethylglyoxime–urethane groups exist in the polymer backbone rather than the cross-linking points, so that the networks can be reorganised more

**Fig. 4 Pattern diagram and paper-cut simulation of sciatic nerve coaptation, gait analysis of sciatic nerve injury model, Masson trichrome staining of gastrocnemius, immunofluorescence staining of regenerated nerve. a** Pattern diagram shows the grouping of sciatic nerve coaptation. **b** Paper-cut simulations of sciatic nerve coaptation. **c** Statistical histogram of coaptation time during the operation ($n = 6$ in each group). **d** Image results of gait analysis in SHE group and suture group. The green blot in the solid yellow box is the footprint of the left lower limb (injury side) of the rat. **e** Statistical histogram of sciatic nerve function index ($n = 5$ in each group). **f** Masson trichrome staining on gastrocnemius in each group. Red part of staining is muscle fibres while blue part is collagen in muscle, scale bar = 100 μm. **g** Statistical plot of the average diameter of gastrocnemius fibres in each group ($n = 5$ in each group). **h** Statistical plot of muscle weight ratio (ipsilateral to contralateral) in each group ($n = 5$ in each group). **i** Hematoxylin–eosin staining of distal anastomotic sciatic nerve. The capillaries inside the nerve bundle are pointed by yellow* ($n = 5$ in each group), scale bar = 100 μm. **j** Statistical plot of capillary quantity per section in sham, SHE and suture group ($n = 5$ in each group). **k** Immunofluorescence staining of S100 (red), NF200 (green) and ChAT (red) of distal anastomotic sciatic nerve in SHE group and suture group, scale bar = 100 μm. **l** through **m** Statistical plot of S100 and NF200 expression, respectively, per section in sham, SHE and suture group ($n = 5$ in each group). **n** Statistical plot of axon quantity per section ($/mm^2$) in sham, SHE and suture group ($n = 5$ in each group). Data were presented as mean ± s.d. Two-tailed unpaired $t$ test (**c**, **e**) was used for comparing anastomosis time and sciatic nerve function index (SFI) between SHE and Suture group. Anastomosis time (**c**): SHE group compared to Suture group, ****$p < 0.0001$; SFI (**e**): SHE group compared to Suture group, ****$p < 0.0001$. Ordinary one-way ANOVA test with Tukey's multiple comparisons test (**j**, **l**, **g**, **h**, **m**, **n**) was used for analysis of capillary quantity, S100 expression, NF200 expression, muscle fibre diameter and muscle weight ratio. Capillary quantity (**j**): Sham compared to SHE, ns $p = 0.9608$; Sham compared to Suture, ns $p = 0.9032$; SHE compared to Suture, ns $p = 0.9857$; Diameter of muscle fibres (**g**): Sham compared to SHE, ns $p = 0.6865$; Sham compared to Suture, ns $p = 0.6462$; Sham compared to Defect, ****$p < 0.0001$; Sham compared to Suture, ns $p = 0.1405$; SHE and Suture compared to Defect, ****$p < 0.0001$; Muscle weight ratio (**h**): Sham compared to SHE, **$p = 0.0015$; Sham compared to Suture, ****$p < 0.0001$; Sham, SHE and Suture compared to Defect, ****$p < 0.0001$; Expression of S100 (**l**): Sham compared to SHE, ns $p = 0.4356$; Sham compared to Suture, ***$p = 0.0007$; SHE compared to Suture, **$p = 0.0065$; Expression of NF200 (**m**): Sham compared to SHE, ns $p = 0.6702$; Sham compared to Suture, **$p = 0.0078$; SHE compared to Suture, *$p = 0.0369$; Axon quantity (**n**): Sham compared SHE, ns $p = 0.0598$; Sham compared to Suture, **$p = 0.0018$; SHE compared to Suture, ***$p = 0.0002$. Source data are provided as a Source Data file. NS no significance, PCL-conduit polycaprolactone conduit.

easily. Accordingly, the SHEs could spontaneously self-heal at room temperature without external stimuli (such as heat and UV light). This feature is favourable for in vivo applications (Supplementary Movie 1). After the SHEs were completely cut to size and manually spliced together at room temperature, the tensile strength after 5 min of healing was already sufficient for certain applications (kPa scale for softer tissue and MPa scale for hard tissue) in vivo (Fig. 2c, f and g). The self-healing efficiencies of SHE0.2, SHE0.5 and SHE1 were over 80% (89 ± 16% for SHE0.2, 85 ± 9% for SHE0.5, 94 ± 5% for SHE1). When the degree of crosslinking increased, the $T_g$ value (20.6 °C) of SHE2 was close to the ambient temperature, which makes the chain movement difficult, resulting in relatively low self-healing efficiency (37 ± 5% for SHE2) (Supplementary Fig. 2). To achieve diverse bionic mechanical properties, SHEs with tunable mechanical properties could be readily adjusted by controlling the crosslinking degree. Because SHE0 is a linear structure, the elasticity was difficult to recover after cyclic tensile testing (Fig. 2h). After introducing crosslinking points, the SHEs (SHE0.2–2) showed elastic recovery because of the crosslinking network, and the elastic recovery increased as the crosslinking degree increased. Notably, the tensile strength of the SHEs varies over a wide range (33 kPa–4.383 MPa), and Young's moduli ranged from 172 kPa to 3.742 MPa (Fig. 2d and e). The SHEs also exhibited similar mechanical properties in the wet state (Supplementary Fig. 5B). Such a wide mechanical tunability can satisfy the requirements for different in vivo applications.

In summary, SHEs have been proven to be autonomously healable, mechanically reliable and biofavourable functional materials. This is the first study to use bio-compatible and biodegradable autonomous SHEs to address practical clinical issues in vivo. To explore the possible application of SHEs in mechanical support, we tentatively used three crosslinked SHEs (SHE0.5, SHE1 and SHE2) to limit aneurysm expansion. We chose PPE adventitial application to establish aneurysm model which have similar pathological changes of aneurysm with human[37,38]. We wrapped the pretrimed SHEs around the aneurysm and achieved in situ seamless sleeving (Supplementary Movie 2). Progressive fibrosis around SHEs will theoretically protect aneurysms from the expansion and rupture after degradation, achieving the purpose of limitation and elimination of residual foreign bodies[39]. SHE0.5 exhibited a better limitation effect for aneurysms than did the SHE1, while the SHE2 unexpectedly caused vessel injury in the presence of collateral circulation instead. The reason for the varied limiting effects can be explained by mechanical tissue matching. Young's modulus of SHE0.5 (1166 ± 198 kPa) is similar to that of the normal aorta of mice (average elastic modulus of a normal mouse aorta: 1000 kPa[40,41], implying that the SHEs can be elastically adapted to a pulsing aorta, achieving the status of best mechanical matching[42]. In contrast, the higher Young's modulus of SHE2 means stiffer than the aorta, leading to repeated injury by mechanical mismatch. Moreover, the decrease in the elastin degradation index and the increase in eNOS expression revealed that SHE wrapping corrected and improved the haemodynamics of aneurysm[43]. SHEs external wrapping also alleviate the inflammation and apoptosis of vascular smooth muscle cell in aneurysm, and reduced the neovascularization level by extra mechanical support and haemodynamic correction[44,45]. Additionally, the limitation efficiency of SHE wrapping on established aortic aneurysms was satisfactory. Compared to silicone rubber which was used in aneurysm wrapping in previous literature[46], the effect against aneurysm progress was slightly superior in SHEs than silicone rubber even with similar mechanical strength. Moreover, due to the self-healing properties, the SHEs can be real-time shaped into an intact structure based on the shape and the branch distribution of the blood vessel during the operation.

Besides, because the surgical suture process can be avoided by using SHEs, some surgical wrapping procedure especially in a narrow surgical field like endoscopic operation or intracranial operation can be greatly simplified, improving the efficiency of operations and reducing the surgical trauma on adjacent tissues. By providing tissue-matched mechanical support and stitch-less operation procedure, these SHEs provide a new promising aneurysm therapy.

Considering the inspiring effectiveness of aneurysm treatment, we tried to extend the SHE application range in vivo. Traditional suturing coaptation is accepted as the standard treatment for peripheral nerve injury. However, there are certain drawbacks of suturing, as mentioned above, causing a substantial decrease in the recovery efficiency. Previous work has reported the effect of several therapies in improving nerve suturing coaptation. Menovsky, for example, tried to evaluate the coaptation efficacy of $CO_2$ laser and fibrin glue treatments relative to sutures. However, they found no difference in functional recovery among these methods[47]. Hence, we pioneered the utilisation of the rapid self-healability and tunable mechanics of SHEs to improve nerve coaptation. Based on the previous studies of nerve regeneration, we originally explored the "LEGO-playing" nerve coaptation protocol (the operation process is shown in Supplementary Fig. 21B and Supplementary Movie 3). Generally, Young's modulus of neural tissues ranges from 100 to 1500 kPa[48]. If the nerve implants such as the PCL exhibit high moduli, corresponding to rigid to neural tissues and resulting in material tissue mismatching, the surgical insertion will trigger both acute and long-term tissue responses[49,50]. Therefore, we reduced the degree of crosslinking and fabricated SHE0.2 to avoid mechanical mismatching between material and nerve. Of note, the time required for coaptation operation with SHE was shortened by 3/4 compared to that required for suturing (Fig. 4c). The magnitude of the time difference is small at a scale of 9 min for single peripheral nerve coaptation. However, the scale of time saving by the use of SHE may be significant in some special clinical situations such as multiple peripheral nerve damage in cases of extensive limb injury or even limb mutilation. Furthermore, according to the encouraging histology and gait analysis results, the recovery of nerves was more efficient than that in the suture, PCL conduit and fibrin-glue. The SFI, calculated from gait analysis, is a common indicator for the evaluation of nerve recovery progress. In the SHE0.2 group, the SFI returned to approximately −30 in only 6 weeks. However, the SFI of simple sutures generally returns to approximately −50 for more than 12 weeks[47,51,52]. Regarding the nerve recovery efficacy in function, the SHE0.2 group was superior to the suture, PCL conduit and fibrin-glue groups. SHE0.2 did not directly stitch the nerves together tightly. The nerve stumps were in a tension-free status, which allowed space between the ends for axon crawling growth, avoided human-made twists and reduced the likelihood of fasciculus misconnection. Moreover, the SHE with a lower Young's modulus than that of the PCL conduit reduced the mechanical stimuli to the sciatic nerve, providing favourable mechanical matching and conduit structure for nerve end protection away from tissue invasion, reducing the apoptosis of Schwann cell[53–56]. In addition, sufficient mechanics provided by SHEs can prevent the extrusion or contraction induced by surrounding limb muscles, while gel-like fibrin-glue alone provided much less mechanical support than the SHEs did[57,58], which might explain the better result of nerve function in the SHE group than in the fibrin-glue group. By seamless LEGO-playing coaptation protocol, the SHE provided a nerve regeneration chamber and material-tissue-matched mechanical protection, reducing the Schwann cell apoptosis in the process of nerve regeneration, and improving the recovery of peripheral nerves after severed injury. Specifically, it should be emphasised that the rapid self-healing property actualises the LEGO-playing coaptation protocol which

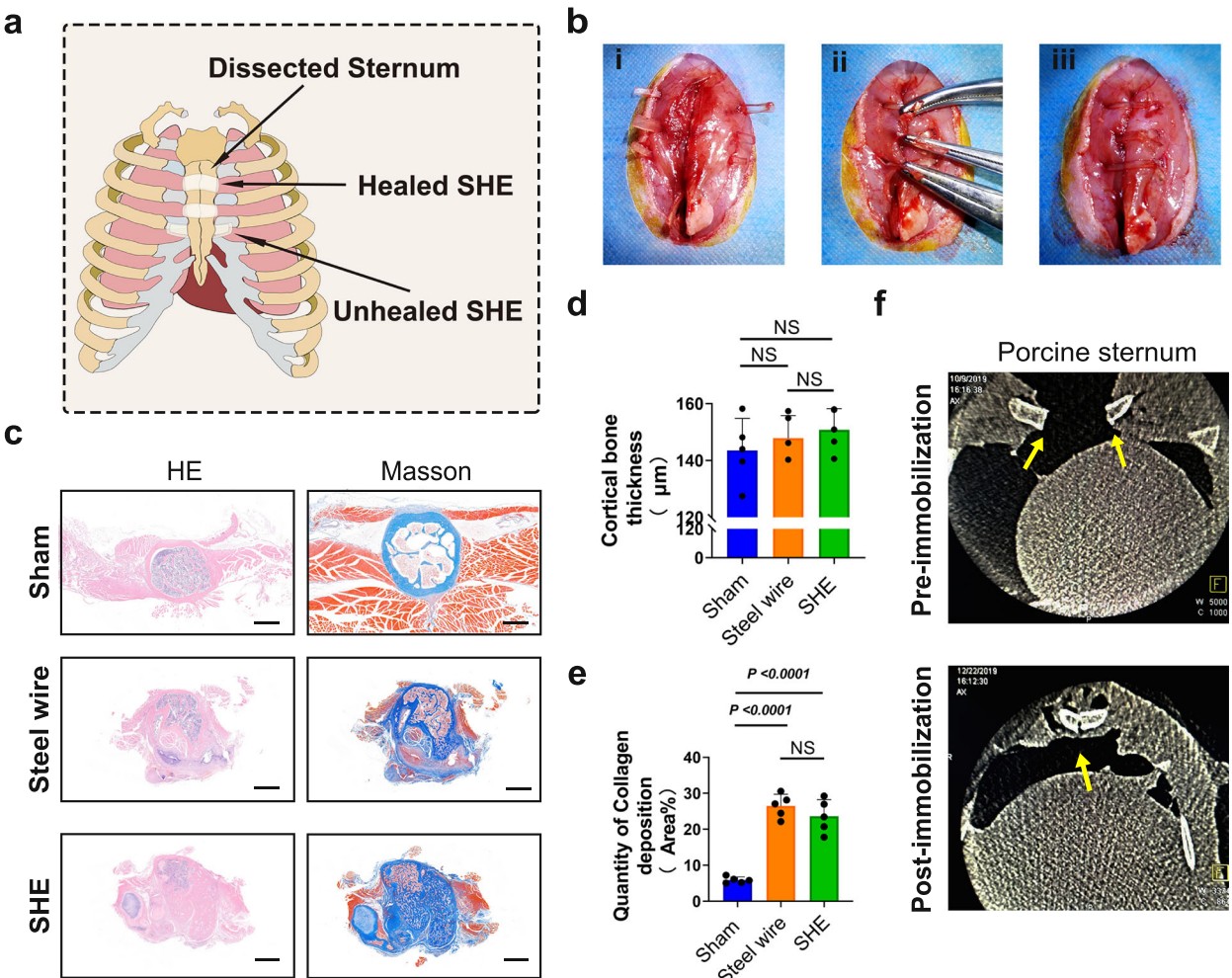

**Fig. 5 Pattern diagram of sternum immobilisation by elastomer, surgical procedure of immobilisation, HE and Masson trichrome staining of sternum, spinal X-ray imagination of porcine sternum. a** Pattern diagram of sternum immobilisation by elastomer. **b** Surgical procedure of sternum immobilisation by elastomer in rat. **c** HE staining of rat sternum in sham, control and elastomer group. Masson staining of rat sternum in sham, steel wire and SHE group (blue part represented the deposition of collagen in sternum bone), scale bar = 500 μm. **d** Statistical scatter diagram of cortical bone thickness of rat sternum in sham, control, elastomer group ($n = 5$ in each group). **e** Statistical scatter diagram of collagen deposition of rat sternum in sham, control, elastomer group ($n = 5$ in each group). **f** Spinal X-ray imagination of porcine sternum (cross section) before and after immobilisation, respectively. High density shadows of sternum were marked with yellow arrows. Data were presented as mean ± s.d. Ordinary one-way ANOVA test with Tukey's multiple comparisons test (**d**, **e**) was used for analysis of cortical bone thickness and quantity of collagen deposition. Cortical bone thickness (**d**): Sham compared to Steel wire, ns $p = 0.7406$; Sham compared to SHE, ns $p = 0.4386$; Steel wire compared to SHE, ns $p = 0.8631$; Quantity of collagen deposition (**e**): Sham compared to Steel wire, ****$p < 0.0001$; Sham compared to SHE, ****$p < 0.0001$; Steel wire compared to SHE, ns $p = 0.3908$. Source data are provided as a Source Data file. NS no significance.

cannot be implemented by other non-self-healing materials, greatly simplifying the procedure of peripheral nerve coaptation, reducing the time of peripheral nerve coaptation, avoiding chronic nerve compression and shortening the learning curve of young neurosurgeons.

The common area between vessels and nerves is soft tissue, which has lower mechanical strength. To fully exploit the application potential of SHEs, we utilised SHE2, which is characterised by a high Young's modulus and good mechanical strength to hard tissue and sternal fracture, which usually occurs after thoracotomy. Steel wire fixation of the sternum is the most popular, but still imperfect, surgical procedure. Nickel, which is present in most medical steel wires, has been reported to be involved in hypersensitivity in up to 15% of the general population[59]. Moreover, as foreign matter in the body, residual steel wire will also poses a potential risk factor for nonunion and wound infection. In this study, the pre-trimmed SHEs were guided through the intercostal muscle of the rat sternum, tightening the sternum and allowing the SHE2 to heal together, thus immobilising the sternum without pneumothorax (Supplementary Movie 4). Due to the intrinsic self-healability, material fatigue will theoretically not occur compared to metal wires. The fractured sternums healed together at 6 weeks post-surgery, and Masson trichrome staining of the sternum showed continuous cortical bone. Additionally, the large animal tests were performed. Following the same immobilisation protocol, the fractured sternum of the pigs was fixed together, and the fracture line was

closed, as proven by spiral X-ray imaging post-operation (Supplementary Movies 5 and 6). In addition, according to the degradation evaluation, SHEs will degrade in vivo, avoiding residual of foreign matter. The results were encouraging, showing satisfactory mechanical compatibility between materials and hard tissues and indicating the potential use of SHEs in orthopaedic disease or injury applications. In the future, by engineering SHEs focusing on bionic mechanics, we expect to utilise SHEs to meet more needs in orthopaedic diseases or injuries such as disorders of the tendon calcaneus or intervertebral disc.

In summary, we established a new family of SHEs, and pioneered the use of the self-healing properties of materials for in vivo applications. To the best of our knowledge, SHEs based on unique hybrid dynamic oxime–urethane covalent bonds and hydrogen bonds, to the best of our knowledge, are the first biocompatible and biodegradable autonomous SHEs. Furthermore, the mechanical properties of the SHEs can be readily tuned by controlling the degree of crosslinking. As a highlight of this study, we tailored SHEs specifically for different types of tissue (soft and hard) in vivo and achieved initiatory encouraging initial results by following the principle of material tissue matching. This level of simplicity, efficiency, and versatility is uncommon in biomaterials. This work truly and fully utilised the properties of self-healing as the key to addressing certain clinical diseases and could inspire a wide range of new applications of self-healing materials for use in biomedicine and accelerate the real high-end applications of self-healing materials.

## Methods

**Materials**. Polytetramethylene ether glycol (PTMEG, Mn = ~1000 g mol$^{-1}$), isophorone diisocyanate (IPDI, 99%), 2-isocyanatoethyl methacrylate and dibutyltin dilaurate (DBTDL, 95%) were purchased from Aladdin. Hexane and toluene were purchased from Sinopharm Chemical Reagent Co., Ltd. The commercial silicone rubber used in aneurysm part as a non-self-healing material control was purchased from Shenzhen Hongyejie Technology Co., Ltd. Bovine pancreatic cholesterol esterase (CE) glycerol (99.5%) was purchased from Sigma-Aldrich. Dimethylglyoxime (DMG, 98%) and acetone (99.8%) were purchased from Sinopharm Chemical Reagent. Phosphate-buffered saline (PBS) were purchased from Shanghai YuanYe Biotechnology. All reagents were used as received without further purification unless otherwise noted.

**Animals**. All animals, including 28 ± 2 g C57BL/6 male mice, 260 ± 10 g Sprague-Dawley male rats, and 25 kg Bama miniature male adult pigs, were obtained from the Laboratory Animal Care Facility of Shanghai Jiao Tong University School of Medicine. All the experiments involving animals in this study were followed by the "Guide for the Care and Use of Laboratory Animals" (NIH, Publication No. 85-23) and "The ARRIVE Guidelines"[60]. Animals were housed at a constant temperature (21 °C) and humidity (35%) under a 12-h light/dark cycle and given standard lab chow and water ad libitum. All animal experiments were approved by the Animal Experimental Ethics Committee of Ruijin Hospital, Shanghai Jiao Tong University School of Medicine and the protocols related to the experimental animals in this study were compliant with specific ethical regulations. The animals were randomised between groups via random number generation by software of Microsoft Excel. We preformed sample size calculations on the online Experimental Design Assistant platform (EDA, National Centre for the Replacement, Refinement & reduction of Animals in Research, https://eda.nc3rs.org.uk/about)[61–63]. Outcome assessors were blinded to the group allocation.

**Synthesis of compound a**. DMG (1.16 g, 10 mmol), 2-methacryloyloxyethyl isocyanate (3.41 g, 22 mmol) and DBTDL (0.05 g) were added to a three-necked flask containing 15 mL of toluene, and stirred at 60 °C for 12 h under a nitrogen atmosphere. After cooling to room temperature, the reaction solution was then poured into 200 mL of hexane, and a white solid was precipitated. The precipitate was filtered and washed with 200 mL of hexane and dried under reduced pressure for 5 h at room temperature to yield a white powder.

**Synthesis of SHEs**. Based on the increase of glycerol molar ratio in SHEs synthesis, they were named SHE0, SHE0.2, SHE0.5, SHE1, SHE2 in sequence (SHEX indicates that the molar ratio of glycerol/PTMEG is X/4). PTMEG (4 g, 4 mmol), DMG (0.464 g, 4 mmol), glycerol (SHE0: 0 g, 0 mmol; SHE0.2: 0.0184 g; 0.2 mmol; SHE0.5: 0.046 g, 0.5 mmol; SHE1: 0.092 g, 1 mmol; SHE2: 0.184 g; 2 mmol) and IPDI (SHE0: 1.776 g, 8 mmol; SHE0.2: 1.8426 g; 8.3 mmol; SHE0.5: 1.9425 g, 8.75 mmol; SHE1: 2.109 g, 9.5 mmol; SHE2: 2.442 g; 11 mmol) were dissolved in 5 mL of acetone in a glass vessel equipped with a magnetic stirrer at 50 °C for 20 h. Then, the reaction mixture was poured into a polytetrafluoroethylene mould at 90 °C for 2 h before the solvent was evaporated under solvent with vacuum at 90 °C for 2 min. The mixture reacted at 80 °C for 20 h before being further cured with vacuum at 75 °C for another 10 h to produce SHEs (catalyst free). In the preparation of c-SHEs (elastomers with a catalyst), the catalyst DBTDL (0.01 g) was added to the reaction mixture immediately after the addition of IPDI, and the remainder of the process was the same as that used for SHE1.

**Preparation of PCL conduits**. A non-healable polymer catheter was prepared by coating a 1.2 mm diameter polytetrafluoroethylene rod with PCL solution (0.1 g/mL, tetrahydrofuran) and volatilising the solvent until the thickness of the tube wall reached 1 mm.

**General characterisation of SHEs**. All tests were performed at room temperature unless otherwise noted. $^1$H NMR spectra were recorded on a Bruker AVANCE III 600 MHz NMR. For the model hydrolysis reaction, compound **a** and water were mixed and dissolved in DMSO-d6. The mixture was monitored by $^1$H NMR at room temperature. Attenuated total reflectance Fourier transform infra-red (ATR-FTIR) spectra were recorded on a ThermoFisher Scientific Nicolet 8700 spectrometer with an ATR accessory. Tensile tests were performed on an electronic universal testing machine (MTS Exceed E42). Rectangular strips (1 mm × 3 mm × 20 mm) cut from large sheets were used. The deflection rates of the uniaxial tensile measurements were 50 and 10 mm min$^{-1}$ for simple tensile and cyclic tensile tests, respectively. The tensile strength was the maximum value determined from the stress–strain curve. The Young's modulus was determined from the slope of the stress–strain curve at 5% of strain. Six specimens were tested and averaged for each sample. The mechanical tests of the original specimens were carried out directly. Restoration of the mechanical properties was assessed by splicing the two individual cut specimens after a pre-determined time at room temperature. For the mechanical test in the wet state, the specimens were completely immersed in physiological saline for 2 h immediately before testing. Differential scanning calorimetry (DSC) was performed on a TA-Q20 differential scanning calorimeter. Samples were heated from 25 to 100 °C, cooled to −80 °C, and reheated to 100 °C at a rate of 10 °C min$^{-1}$ under a nitrogen atmosphere.

All the data were obtained from the second heating curves. The glass transition temperatures were determined by dynamic thermodynamic analysis on a DMA1 (Mettler Toledo) dynamic mechanical analyser. Rectangular samples (ca. 1 mm ($T$) × 3 mm ($W$) × 8 mm ($L$)) were tested at a frequency of 1 Hz and a strain of 0.1%. Heating ramps of 5 °C min$^{-1}$ were applied from −110 to 50 °C. $T_g$ values were calculated from the maximum value of tan Delta. The rheological properties were measured by Discovery HR-2 TA rheometer. Each cylindrical sample tailored to 0.8 cm in diameter by 1 mm thickness was tested within the linear strain range (0.1%). Frequency sweep experiments were performed over an angular frequency $\omega$ range of 0.1–100 rad s$^{-1}$ at 25 and 37 °C for each sample. The air-water contact angles of the samples were measured with a water contact angle instrument (Contact Angle System OCA40, Dataphysics Co., Germany) at room temperature. An in vitro enzymatic degradation test was performed in PBS containing CE at an activity of 200 U mL$^{-1}$ (mimicking the environment of the active enzymatic reaction in vivo). Cuboid-shaped specimens (1 mm ($T$) × 2 mm ($W$) × 4 mm ($L$)) were weighed and immersed in 1 mL of PBS with CE and then incubated at 37 °C. The specimens were retrieved, washed with distilled water, and dried every 36 h. The degree of degradation was determined by the dry-weight change.

**Evaluation of SHEs biocompatibility in vitro**. Considering that the direct contact between the target tissue and SHEs occurs at the outer membrane composed of fibroblasts, the primary mouse areolar fibroblasts were used in this part of the study. Two kinds of elastomers were used: an elastomer with a catalyst that we previously synthesised as a negative control and SHE2. In addition, PCL, which was approved by the FDA for medical apparatus fabrication[28], was used as a positive control. The Cell Counting Kit-8 (CCK-8) (Dojindo Laboratories, Kumamoto, Japan) was used to verify the biocompatibility of materials at 6, 12, 24, 48, and 72 h post seeding. Live/dead cell staining (04511, Sigma-Aldrich) was also performed 24, 48, and 72 h post seeding. More details about these experiments are provided in supplemental file.

**In vivo degradation of SHEs and evaluation of SHE sides effects**. The synthesised SHEs were dried and weighed before being implanted subcutaneously on the backs of C57BL/6J mice. c-SHEs was used as a negative control in Ly6G analysis (acute inflammation analysis) and PCL was used as a positive control in F4/80 analysis (chronic inflammation analysis). More details regarding these experiments are described in supplemental file.

**Animal experimental design**. After analysis of their mechanical properties and biocompatibility, the SHEs were prepared for utilisation in three disease models, namely, an aneurysm model, a sciatic nerve defective model and a sternotomy model in vivo, following the principles of tunable mechanics. The animal experimental designs were as follows: (1) for the aneurysms models, animals (C57BL/6 mice) were divided into eight groups, i.e., the sham, aneurysm (as a negative control), SHE2, SHE1, SHE0.5, non-self-healing material (silicone rubber), SHE1 on a 3-day-established aneurysm (eSHE1) and SHE0.5 on a 3-day-established aneurysm (eSHE0.5) groups; (2) for the sciatic nerve model, animals (SD rats) were divided into six groups, i.e., the sham, sciatic nerve defect (as a negative control), sciatic nerve suture, fibrin-glue group, PCL-conduit and SHE0.2 groups; (3) for the sternum immobilisation model, animals (SD rat) were divided into three groups, i.e., the sham group, steel wire group (as a positive control) and SHE2 groups; Additionally, the pigs were utilised for this experiment.

**Limiting the progression of an abdominal aortic aneurysm and evaluation of the corresponding result**. C57BL/6 male mice (28 ± 2 g) were used in this part. The animals were randomly divided including: the sham group, aneurysm group, SHE2 group, SHE1 and SHE0.5 group. Besides, to compare the effect against aneurysm between SHE and other non-self-healing material, we added silicone rubber which possesses the similar mechanical strength with SHE as non-self-healing control group. The topical application of porcine pancreatic elastase (E1250, Sigma-Aldrich) onto the abdominal aorta of C57BL/6 mice was performed to mimic the progression of an arterial aneurysm as previously reported[37]. Moreover, to further verify the efficiency of SHEs in aneurysm limitation, we also applied the SHE1 and SHE0.5 in a 3-day-established aneurysm models (eSHE1 and eSHE0.5 group). In vivo images of the enlarged aneurysms were captured before the wrapping operation. The details of the operation are described in the supplemental file, and the video recorded by a camera is provided in supplemental movie. On 14-day post aneurysm establishment, all animals underwent MRI scanning, and then arterial specimens were obtained for subsequent analysis. The details of MRI scanning are described in the supplemental file. The aorta diameter was determined from the MRI image. MRI measurements were performed by two assessors who measured the maximum radial lines of the lumen as the inner aortic diameter using Sante MRI Viewer software. Repeatability was analysed and assessed with the Bland–Altman difference[64]. The target aorta was harvested in each group for further histological analysis, including haematoxylin–eosin (HE), elastica van Gieson (EVG), immunohistochemical (IH) and IF staining. Image J was used for calculating positive area percent. Details of the histological analysis are provided in the supplemental file. The q-PCR was performed to analysis the transcription foldchange of eNOS and caspase-3 mRNA of aorta tissue. The details of the PCR and primers name and sequence can be seen in supplementary file and table.

**Sciatic nerve coaptation and assessment of the corresponding result**. For the peripheral nerve model, SD male rats ($260 \pm 10$ g) were used in this experiment. The animals were divided into six groups: sham group, defect group, suture group, fibrin-glue group, PCL conduit group and SHE0.2 group. For the fibrin-glue, we chose the commercial product- FIBRINGLURAAS® fibrin sealant (Nuance Biotech Inc., China). Details of the nerve operations in the sham, defect, suture, fibrin-glue and PCL conduit groups are described in the supplemental file. In the SHE0.2 group, after amputation, the proximal nerve and distal nerve were fixed on the muscle by suturing the adventitia of the nerve with one stitch, respectively. The aim of fixing the nerve was to avoid axial shifting along the long axis after SHE0.2 intervention, and this process also reduced the tension between the stumps. After fixation, the pretrimmed SHEs were wrapped around the two ends of the nerve and allowed to heal, bridging the junction of the nerve with the presence of a small gap between the two ends of the nerve. Details of the operation in the material group can be viewed in the supplemental movie (Movie 3). To estimate the function of the regenerated nerves after nerve coaptation, an evaluation was performed by means of the CatWalk XT system (Noldus, Wageningen, Netherlands) and calculated using the static sciatic index[65]. The sciatic nerve function index (SFI) was calculated by the following equation:

$$\text{SFI} = -38.3 \times \frac{(\text{EPL} - \text{NPL})}{\text{NPL}} + 109.5 \times \frac{(\text{ETS} - \text{NTS})}{\text{NTS}} + 13.3 \times \frac{(\text{EIT} - \text{NIT})}{\text{NIT}} - 8.8 \tag{1}$$

E and N indicate the experimental side and contralateral normal side, respectively on each rat. The abbreviations of PL, TS and IT indicate the lengths from the third toe to the heel (PL), the first to the fifth toe (TS) and the second toe to the fourth toe (IT), respectively.

An SFI of approximately 0 indicates normal or recovered nerve function, while an SFI of approximately $-100$ represents total nerve dysfunction. The target nerve was harvested in each group on the week 6 post coaptation for further histological analysis, including haematoxylin–eosin (HE), fluorogold (FG) staining, Luxol Fast Blue (LFB) staining, transmission electron microscopy (TEM), Masson trichrome (MT) staining and IF staining. Image J was used for calculating positive area percent. The details of the histological analysis are provided in the supplemental file. The q-PCR was performed to analysis the transcription foldchange of caspase-3 mRNA of coaptated nerve tissue. The details of the PCR and primers name and sequence can be seen in supplementary file and table.

**Sternum immobilisation and assessment of the corresponding result**. To explore the possibility of replacing traditional steel wire retainers, median sternotomy was performed on SD rats after routine anaesthesia and orotracheal intubation. In the SHE2 group, the SHE2s (2 mm × 2 mm × 20 mm), which were shaped into strips prior to the procedure, were traversed through intercostal muscles on both sides, and the two ends of the SHE were fixed and healed together to achieve sternum closure. In the steel wire group, the sternal incision was sutured by a stainless-steel wire with a diameter of 0.2 mm. At 6 weeks post-surgery, C-arm-X-ray in spinal bone mode was chosen to image the healed sternum clearly, and then, the sternums were harvested for subsequent analysis. Additionally, in this experiment, a porcine model (male adult Bama miniature pig, 25 kg) was also utilised to probe the effect of the SHE2 (10 mm × 10 mm × 150 mm). Operations on these large animals were performed on a hybrid surgery platform, and the details of the operation are provided in the supplemental file.

**Statistical analysis**. All quantitative results are presented as the mean ± standard deviation. GraphPad Prism 8.0 software (GraphPad Software, San Diego, CA, USA) and SPSS 22 (IBM, Armonk, NY, USA) statistics software were used for statistical analysis. The data were evaluated using single-factor analysis of variance (one-way-ANOVA) and Tukey's test for simultaneous paired comparisons if conditions of normality and equal variance were met. If the sample data were tested assuming unequal variance, data between two groups were compared with nonparametric tests using the Mann–Whitney $U$ test and data among three or more groups were compared with nonparametric tests using the Kruskal–Wallis test[62]. A value of $p < 0.05$ was considered statistically significant.

**Reporting summary**. Further information on research design is available in the Nature Research Reporting Summary linked to this article.

## Data availability

All relevant data are available from the corresponding author upon reasonable request. All the data supporting the findings of this study are available within this article, supplementary information files. Source data are provided with this paper.

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

## Acknowledgements

The authors would like to be grateful to all the animals in this study for their sacrifice to promote the development of human biomedical enterprises. We thank Dr. Ming Qi who is affiliated in Department of Nuclear Medicine, Fudan University Shanghai Cancer Center and Center for Biomedical Imaging, Fudan University for his professional and kind help to acquire the MRI image of mice aorta. We thank Professor Qian Huang who is affiliated in Institutes of Brain Science, Fudan University for her professional support of rat gait analysis. We also thank laboratory technician Peng Ke for his patient help to prepare the specimens of animals. We gratefully acknowledge the financial supports by the Natural Science Foundation of China (81671832, 81571826), the Shanghai Municipal Education Commission-Gaofeng Clinical Medicine Grant Support (826158), the National Key Research and Development Programmes for Acute Aortic Syndrome in High-risk Early-warning and Intervention (2017YFC1308000), the Natural Science Foundation of Shanghai (18ZR1401900), Shanghai Belt and Road Joint Laboratory of Advanced Fiber and Low-dimension Materials (Donghua University) (18520750400), the Fundamental Research Funds for the Central Universities, DHU Distinguished Young Professor Programme (LZA2019001).

## Author contributions

C.J., L.Z., Q.Z., Z.Y. and X.Y. designed the study. Q.Z., Z.Y. and X.Y. supervised the whole work. C.J. performed the biomedical experiments. L.Z. and Z.Y. designed and fabricated the self-healing elastomer. L.Z., Z.L. and Q.G. performed the analysis of characterisations of materials. Q.Y., B.Q. and H.S. analysed the data of aneurysm part. M.L. and Q.L. analysed the data of peripheral nerve part. R.Y. and S.H. analysed the data of sternum part. C.J., L.Z., Q.Z., Z.Y. and X.Y. co-wrote the manuscript. All results were discussed with all authors.

## Competing interests

The authors declare no competing interests.
