## [Peer Review File · Nature Communications]

REVIEWER COMMENTS

Reviewer #1 (Remarks to the Author):

This interesting study investigated the effect of self healing elastomers on a number of pathological problems relevant to human disease. The findings are of great interest.

I have focused on the aneurysm part as advised by the editor.

The current aneurysm study is focused on 20 mice included in four groups. This would be substantially improved by:

- providing more methodological detail to the presentation as per the ARRIVE criteria and as expected of clinical studies. How exactly was aortic diameter measured on MRI and morphology? What was the repeatability of diameter measurement? Were mice randomised between groups and if so how? Were outcome assessors blinded to group allocation and how was that maintained? How was the small sample size justified? What statistical tests have been used to compare between these small numbers of animals? Parametric tests would seem inappropriate.
- Please provide MRI and morphometry pictures for all aortas not just selected examples
- In clinical practice a treatment is needed to limit AAA growth. The investigation would be greatly strengthened if a study focused on the effect of SHE in established aneurysms (ie aneurysms that are confirmed to be present through imaging first before administering SHEs) was added and in a much larger group of mice in order to confirm the pioneering effect claimed.

Jonathan Golledge

Reviewer #2 (Remarks to the Author):

In this manuscript, the authors developed a series of biocompatible, biodegradable and self-healing elastomers (SHEs) for 3 in vivo applications, including aneurysm, nerve anastomosis and bone immobilization. The authors find that the biodegradability and self-healing ability of the elastomer are attributable to the dynamic oxime-urethane covalent bonds and hydrogen bonds, and its mechanical properties can be tuned by controlling the crosslinking degree. They demonstrated the therapeutic efficiency of these materials in three animal models.

The manuscript is interesting, well designed and prepared, it should be acceptable after addressing the following issues:

1. The title should be more specific and meaningful
2. The introduction should be re-organized. What is the challenge of the material for self-healing in vivo, compared with in vitro at physiological temperature? There are considerable previous reports of materials that enable autonomous self-healing at physiological temperature, so the mechanism about the challenge of self-healing in vivo should be clarified. In addition, listing the introduction of the three diseases may not be suitable. A general description may be better.
3. No clear peak can be found on the DSC curves in Fig S2. The authors explain that the glass transition temperature is below -60°C , so the DSC test with lower temperatures or the DMA test should be added.
4. More material characterizations, such as rheological behavior, XPS, SEM or AFM, should be provided.
5. The crosslinking structure can reduce the flexibility of polymer chain, thereby influence the self-healing performance. There should be a tradeoff between mechanical strength and self-healing property. How to ensure both the healing efficiency and toughness of this elastomer? What's the healing efficiency of three crosslinking degrees of elastomers? More explanations should be added.
6. DBTDL (dibutyltin dilaurate) catalyst is toxic. Is DBTDL thoroughly removed after the reaction? The release of DBTDL should be considered during degradation.

Reviewer #3 (Remarks to the Author):

Re: NCOMMS-20-23626

Summary:

In this manuscript, Jiang and coauthors present applications of self healing elastomers in a variety of models of clinical conditions. They describe the synthesis of the polymers and their characterization, and they report biocompatibility data.

The authors performed an acute sciatic nerve cut and repair using either 10-0 sutures or with a single 10-0 suture plus SHE0.2, and included a sham control as well as an unreconstructed defect control. They assessed walking tracks and SFI, as well as neurofilament staining. They report shorter suture time, and improved SFI at six weeks postop.

Comments:

As my expertise lies in nerve regeneration I will confine my comments to those involving the nerve components of the work.

Overall I found the work to be novel and interesting.

1. I understand that the authors oriented the sciatic nerve in the SHE0.2 groups to prevent rotational misalignment, but it is not stated whether this was also done in the suture group and this should be clarified. If the nerve coaptations in the suture group were anatomically misaligned then that would account for many of the differences reported in the outcome metrics.
2. I believe the claim that the technique is faster than suturing, but I would add that the magnitude of this difference is small in terms of actual minutes.
3. The authors do not show the time course of the SFI data (such as preop, short term following surgery, and other time points).
4. The n is not stated.
5. Panel K and L show greater NF200 expression, but histomorphometry (apart from axon density, which is derived and not measured directly) is not reported, nor is retrograde labeling of neurons that regenerated their axons. It is not clear how many biological replicates were used for each metric, as the n varies between panels.
6. Given the popularity of fibrin glue polymers in nerve surgery I would like to see a comparison between cut and repair with fibrin glue vs. SHE, since that would represent the main source of competition in this clinical space (not sutures, necessarily). I suspect that there would be no difference in metrics of nerve regeneration in such an experiment.

Reviewer #4 (Remarks to the Author):

First the authors should be congratulated for putting together a very broad clinically oriented set of in vivo experiments examining the self healing elastomers they have developed.

From my review I have three concerns.

1. Grammar and syntax are poor and in some cases make it difficult to follow the logic of the discussion.

2. The self healing elastomers they describe are not novel, and there is an implication that they are. They were well described in Advanced Materials in 2019 for instance.

A Highly Efficient Self-Healing Elastomer with Unprecedented Mechanical Properties

<https://doi.org/10.1002/adma.201901402> describes the very same chemistry that has been presented.

3. I believe an important study group is missing. Many biomaterials are conductive for tissue and a control group of a non 'self healing biomaterial' should have been used. For instance, it is well known and studied that polymeric nerve conduits can have beneficial effects on nerve conduction and healing. Why were they not used as controls? Without that group it is not clear what the real benefit of these polymeric materials are.

Dear Reviewers:

Thank you for your comments concerning our manuscript “The Pioneering Applications of Self-healing Elastomer *in vivo*” (ID: NCOMMS-20-23626). We have carefully studied these comments which are all valuable and very helpful for revising and improving our paper, as well as the importance guiding significance to our researches. We have made several corrections and added more specific contents in revised manuscript. The content alterations were marked with color of **RED** and grammar corrections were marked with color of **GREEN** in the manuscript and the responses for comments of reviewers were listed orderly in following part of this letter.

I. Summary of main added experiment contents

Part	Added content	Purpose	Response to
Materials characterizations	DMA test	To illustrate material characterizations more specifically	Reviewer 2#
	Rheological test		
	Contact angles		
Aneurysm	SHE wrapping on established aneurysm	To present aneurysm limitation effect of SHE in clinical practice situation such as wrapping on established aneurysm	Reviewer1#
Nerve anastomosis	More time points of SFI were added	To illustrate the alterations of SFI in each group with time variation	Reviewer3#
	Histomorphometry of axons and retrograde labeling of neurons were added	To illustrate axons regeneration situations more specifically	Reviewer3#
	Nerve anastomosis with fibrin-glue	Setting new control groups to compare the anastomosis effect of SHE with other current materials or therapies	Reviewer3#
	Nerve anastomosis with PCL-conduit		Reviewer4#

II. Details of response

1. Original reviewer #1 comments (remarks to the author):

This interesting study investigated the effect of self healing elastomers on a number of pathological problems relevant to human disease. The findings are of great interest.

I have focused on the aneurysm part as advised by the editor.

The current aneurysm study is focused on 20 mice included in four groups.

This would be substantially improved by:

- providing more methodological detail to the presentation as per the ARRIVE criteria and as expected of clinical studies. How exactly was aortic diameter measured on MRI and morphology? What was the repeatability of diameter measurement? Were mice randomised between groups and if so how? Were outcome assessors blinded to group allocation and how was that maintained? How was the small sample size justified? What statistical tests have been used to compare between these small numbers of animals? Parametric tests would seem inappropriate.

- Please provide MRI and morphometry pictures for all aortas not just selected examples

- In clinical practice a treatment is needed to limit AAA growth. The investigation would be greatly strengthened if a study focused on the effect of SHE in established aneurysms (ie aneurysms that are confirmed to be present through imaging first before administering SHEs) was added and in a much larger group of mice in order to confirm the pioneering effect claimed.

Response to reviewer 1#

Comment:

This interesting study investigated the effect of self-healing elastomers on a number of pathological problems relevant to human disease. The findings are of great interest.

Response:

Thank you for your affirmation and reviewing the manuscript in your busy schedule and provided us with your professional comments and suggestions, the questions and the certain responses were presented in following part orderly.

Questions and response:

1. providing more methodological detail to the presentation as per the ARRIVE criteria and as expected of clinical studies. How exactly was aortic diameter measured on MRI and morphology? What was the repeatability of diameter measurement? Were mice randomised between groups and if so how? Were outcome assessors blinded to group allocation and how was that maintained? How was the small sample size justified? What statistical tests have been used to compare between these small numbers of animals? Parametric tests would seem inappropriate.

Response:

Thank you for your questions. The methodological details and several added results were illustrated more specifically in this response letter and revised manuscript.

As to “How exactly was aortic diameter measured on MRI and morphology”, we performed MRI measurement: By flowing void effect of MRI in the cross section, the aorta will be illustrated as high signal region which is the structure of vessel lumen. Then we measured the maximum radial lines of lumen as aortic diameter on *Sante MRI Viewer* software. Morphology measurement: After humanely euthanasia of animals, we injected the fixative solution into the blood vessel from the left ventricle for fixing the vessel in a non-collapsed state. Then we took the blood vessel specimen at the place where the maximum diameter of the vessel can be seen to perform HE staining and measure the maximal lumen diameter of the vessel under microscope. The protocols of MRI and pathological measurement were added in method of revised manuscript at manuscript-page7- line26.

As to “What was the repeatability of diameter measurement?”, two assessors performed the measurements and the diameter of aorta were measured three times in total. Specifically, assessor 1# performed all measurements in a random order and repeated the measurements 1 weeks later to determine intra-observer variability. Assessor 2# performed the measurements once for the interobserver variability analysis. Repeatability was analyzed and assessed with the **Bland-Altman difference** based on previous studies^{1,2}. Intra-assessor repeatability (coefficient of repeatability=0.02689, 95%CI=0.02077–0.03883, CV0%); Inter-assessor repeatability (coefficient of repeatability= 0.05757, 95% CI= 0.04404–0.08314, CV 5%). Intra- and interobserver were plotted as the mean differences ± 1.96 standard deviations. The results of repeatability analysis were acceptable. The detail data of repeatability analysis were presented at Manuscript-Page7- Line28.

The results of repeatability analysis

As to “Were mice randomized between groups and if so how?”, the animals were randomized between groups. The principal of group randomization was followed by previous study³. Prior to performing experiment, we numbered the animals (animals

were homogeneity, male and weight in 25g) and then grouped the animals via method of random number generated by computer (*Microsoft Excel*). Specifically, after numbering the animals (from 1 to 20 using ear studs), we typed those 20 natural numbers into **column A** in *Excel* orderly. Secondly, we used function of random (*fx=RAND()*) to generated 20 different random numbers (between 0 to 1) in **column C**, next we copied those 20 random numbers by text formatting (avoiding generated random number changing once more) in **column B**, then we ranked 20 random numbers of **column B** in ascending order and extended rank scale to column A making sure the numbers in column A following the order of column B, finally we assigned every continuous 5 numbers of column A from top to bottom orderly into the four certain study groups (including sham group, aneurysm group, SHE0.5 group and SHE1 group respectively) and finished group randomization. The method of randomization was briefly added in the revised manuscript-page6-line18.

As to “Were outcome assessors blinded to group allocation and how was that maintained?”, outcome assessors were blinded to group allocation. Specifically: First, allocation concealment: the investigators were unaware of the group to which the next animal taken from a cage will be allocated. Second, blinded conduct of the experiment: animal caretakers were blinded to the allocation sequence. Even surgeons performed the certain operations, which they were aware of the specific allocation sequence, but they did not participate in outcomes assessing and communicate with investigators about allocation situations. Last, blinded assessment of outcome: investigators assessing, measuring or quantifying experimental outcomes were blinded to the intervention. The method of blinding process was briefly added in the revised manuscript-page6-line22.

As to “How was the small sample size justified?”, we performed the sample size justification by the principal of ARRIVE criteria⁴. Based on the previous published research⁵, we performed sample size calculations on online platform-EDA (Experimental Design Assistant, *national centre for the replacement refinement & reduction of animals in research*, <https://eda.nc3rs.org.uk/about>). Following the instructions of EDA, we finished the experiment design diagram and typed the certain parameters which were based on results from published studies for power analysis (certain parameters: 90% power and a two-sided test; anticipated effect size ($m_1 - m_2$): 0.5mm, where m_1 and m_2 represent the mean in treatment and control groups, the “0.5mm” is minimum anticipated interference effect of wrapping between treatment and control groups; previous published aorta diameter SD is 0.21mm⁵), the *n per group* was then calculated as 5, which was also in accordance with the previous study⁶. The strategy of sample size calculation was briefly described in method part of revised manuscript-page6-line19.

As to “What statistical tests have been used to compare between these small numbers of animals? Parametric tests would seem inappropriate.”, all quantitative results

are presented as mean \pm standard deviations. The strategy of statistics analysis was based on previous researches^{7,8}. Briefly, before choosing specific statistical test, we analyzed data property in terms of normality and homogeneity of variances, then we chose the parametric or non-parametric tests based on the property of data. In this study, the data of diameter and expression of collagen were normality and equal variance, hence, we used one-way ANOVA and Tukey's multiple comparisons test to analyze difference among groups. While, for data of expression of eNOS which was not equal variance and data of Elastin grade which was ranked data, we used nonparametric tests as Kruskal-Wallis test to analyze difference among groups. Specially, A value of $P < 0.05$ was considered statistically significant. The strategy of statistics analysis was briefly described in method part of revised manuscript-page9-line13.

2. Please provide MRI and morphometry pictures for all aortas not just selected examples

Response:

All MRI images and morphometry pictures were presented in the response and revised supplement file (Fig.S9, Fig.S10 and Fig.S14).

Morphometry pictures of aortas

Morphometry pictures of aortas in groups of SHEs wrapping on established aneurysm

MRI images of aortas in each group:

- In clinical practice a treatment is needed to limit AAA growth. The investigation would be greatly strengthened if a study focused on the effect of SHE in established aneurysms (ie aneurysms that are confirmed to be present through imaging first before administering SHEs) was added and in a much larger group of mice in order to confirm the pioneering effect claimed.

Response:

Thank you for your suggestion. To strength the claim of SHEs' efficiency on aneurysm limiting, we wrapped the SHEs on established aneurysm models. Based on the previous study⁹, 3rd day post of aneurysm establish is a crucial time point in the ascending phase of aneurysm expansion. According to the preliminary study (Fig. S11), SHE2 presented the mechanical injury to abdominal aorta, hence, we did not utilise SHE2 in wrapping on established aneurysm models. We performed the SHEs (including SHE0.5 and SHE1, and we named new groups as eSHE0.5 and eSHE1 respectively) wrapping on the third day after establishing of aneurysm model. The *in vivo* images of enlarging aneurysm were captured before wrapping operation. On the 14th day post of aneurysm establish, the animals were received MRI scanning to illustrate *in vivo* morphological alterations of abdominal aorta in groups, then the specimens of aorta were extracted from sacrificed animals for more pathology analyses. eSHE1 and eSHE0.5 group showed satisfactory limitation effect on 3-day established aneurysm based on the MRI images before and after wrapping comparison. Also, there is no statistical diameter difference of before and after wrapping in eSHE1 and eSHE0.5 group. The elastin grade was no difference among aneurysm, eSHE1 and eSHE0.5 group. The expressions of eNOS in the eSHE1 and eSHE0.5 group were statistically higher than aneurysm group. Additionally, the mean ratio of collagen I/III in the eSHE1 and eSHE0.5 were no statistical difference. The details of result and discussion were presented at manuscript page11-line25, page11-line31, page12-line6, page12-line10 page12-line15, page12-line24 and page17-line18.

Morphometry and MRI picture of aorta in eSHE0.5 and eSHE1group.

SHE 0.5

SHE 1

MRI picture summary of aorta in eSHE0.5 and eSHE1group.

EVG, eNOS and collagen staining of aorta in established aneurysm wrapping groups.

1. Kamman, A.V., *et al.* Standardized Protocol to Analyze Computed Tomography Imaging of Type B Aortic Dissections. *J Endovasc Ther* **23**, 472-482 (2016).
2. Krishna, S.M., *et al.* Fenofibrate increases high-density lipoprotein and sphingosine 1 phosphate concentrations limiting abdominal aortic aneurysm progression in a mouse model. *Am J Pathol* **181**, 706-718 (2012).
3. Ioannidis, J.P., *et al.* Comparison of evidence of treatment effects in randomized and nonrandomized studies. *Jama* **286**, 821-830 (2001).
4. Kilkeny, C., Browne, W., Cuthill, I.C., Emerson, M. & Altman, D.G. Animal research: reporting in vivo experiments: the ARRIVE guidelines. *Br J Pharmacol* **160**, 1577-1579 (2010).

5. Lareyre, F., *et al.* TGF β (Transforming Growth Factor- β) Blockade Induces a Human-Like Disease in a Nondissecting Mouse Model of Abdominal Aortic Aneurysm. *Arteriosclerosis, thrombosis, and vascular biology* **37**, 2171-2181 (2017).
6. Krishna, S.M., *et al.* Wnt Signaling Pathway Inhibitor Sclerostin Inhibits Angiotensin II-Induced Aortic Aneurysm and Atherosclerosis. *Arteriosclerosis, thrombosis, and vascular biology* **37**, 553-566 (2017).
7. Moran, C.S., Jose, R.J., Biros, E. & Golledge, J. Osteoprotegerin deficiency limits angiotensin II-induced aortic dilatation and rupture in the apolipoprotein E-knockout mouse. *Arteriosclerosis, thrombosis, and vascular biology* **34**, 2609-2616 (2014).
8. Junqueira, C., *et al.* Cytotoxic CD8+ T cells recognize and kill Plasmodium vivax-infected reticulocytes. *Nature Medicine* **24**, 1330-1336 (2018).
9. Maegdefessel, L., *et al.* miR-24 limits aortic vascular inflammation and murine abdominal aneurysm development. *Nat Commun* **5**, 5214 (2014).

2. Original reviewer #2 comments (remarks to the author):

In this manuscript, the authors developed a series of biocompatible, biodegradable and self-healing elastomers (SHEs) for 3 in vivo applications, including aneurysm, nerve anastomosis and bone immobilization. The authors find that the biodegradability and self-healing ability of the elastomer are attributable to the dynamic oxime-urethane covalent bonds and hydrogen bonds, and its mechanical properties can be tuned by controlling the crosslinking degree. They demonstrated the therapeutic efficiency of these materials in three animal models.

The manuscript is interesting, well designed and prepared, it should be acceptable after addressing the following issues:

- 1. The title should be more specific and meaningful*
- 2. The introduction should be re-organized. What is the challenge of the material for self-healing in vivo, compared with in vitro at physiological temperature? There are considerable previous reports of materials that enable autonomous self-healing at physiological temperature, so the mechanism about the challenge of self-healing in vivo should be clarified. In addition, listing the introduction of the three diseases may not be suitable. A general description may be better.*
- 3. No clear peak can be found on the DSC curves in Fig S2. The authors explain that the glass transition temperature is below -60°C, so the DSC test with lower temperatures or the DMA test should be added.*
- 4. More material characterizations, such as rheological behavior, XPS, SEM or AFM, should be provided.*
- 5. The crosslinking structure can reduce the flexibility of polymer chain, thereby influence the self-healing performance. There should be a tradeoff between mechanical strength and self-healing property. How to ensure both the healing efficiency and toughness of this elastomer? What's the healing efficiency of three crosslinking degrees of elastomers? More explanations should be added.*
- 6. DBTDL (dibutyltin dilaurate) catalyst is toxic. Is DBTDL thoroughly removed after the reaction? The release of DBTDL should be considered during degradation.*

Response to reviewer 2#

Comments:

In this manuscript, the authors developed a series of biocompatible, biodegradable and self-healing elastomers (SHEs) for 3 in vivo applications, including aneurysm, nerve anastomosis and bone immobilization. The authors find that the biodegradability and self-healing ability of the elastomer are attributable to the dynamic oxime-urethane covalent bonds and hydrogen bonds, and its mechanical properties can be tuned by controlling the crosslinking degree. They demonstrated the therapeutic efficiency of these materials in three animal models.

The manuscript is interesting, well designed and prepared, it should be acceptable after addressing the following issues:

Response:

Thank you for your affirmation and support for our current research work. Following your instructions for revising the manuscript, All the issues have been addressed one by one as below and the certain responses for your comments were presented in following part. The corrections were also made in revised version manuscript.

Questions and response:

1. The title should be more specific and meaningful.

Response:

Thank you for your suggestion. To illustrated this study specifically and meaningfully, the title of manuscript has been replaced with - ***The Self-healing Polyurethane-elastomer with Mechanical Tunability for Multiple Biomedical Applications in vivo*** -.

2. The introduction should be re-organized. What is the challenge of the material for self-healing in vivo, compared with in vitro at physiological temperature?

There are considerable previous reports of materials that enable autonomous self-healing at physiological temperature, so the mechanism about the challenge of self-healing in vivo should be clarified. In addition, listing the introduction of the three diseases may not be suitable. A general description may be better.

Response:

Thank you for your suggestion. The introduction has been re-organized concisely. Although there have been reports of materials that can self-heal around body temperature, they are difficult to be used in the body. The reasons are illustrated in following contents. First, the time of self-healing process is too long in several reported self-healing materials, which seems inappropriate in surgical operations, since the shorter the operation time, the better the recovery efficiency of the patient. Second, to achieve the rapid self-healing process, several self-healing materials will be added in kinds of catalysts or harmful agents such as cupric ion or other heavy metal ion, which will be harmful to living body. Third, several self-healing materials such as hydrogel are lack of sufficient mechanics, which are hard to provide mechanical support for structure surgical diseases as the diseases described in this study. While several self-healing materials such as silicon-carbide-ceramic-matrix-composite characterized with high elastic modulus will induce mechanical injury for softer tissue as vessel or nerve. Moreover, the degradation properties of materials are also critical, but they are rarely studied previously.

3. No clear peak can be found on the DSC curves in Fig S2. The authors explain that the glass transition temperature is below -60°C, so the DSC test with lower temperatures or the DMA test should be added.

Response:

Thank you for your suggestion. The DSC tests may not be sensitive enough, so we have now conducted DMA tests to reveal that the glass transition temperature of the self-healing elastomers. The glass transition temperature of SHE0, SHE0.2, SHE0.5,

SHE1, SHE2 were -35.7°C , -21.2°C , -18.1°C , -7.2°C and 20.6°C , respectively. The DMA results have been added in supplement file (Fig. S2).

	SHE0	SHE0.2	SHE0.5	SHE1	SHE2
T_g ($^{\circ}\text{C}$)	-35.7	-21.2	-18.1	-7.2	20.6

4. More material characterizations, such as rheological behavior, XPS, SEM or AFM, should be provided.

Response:

Thank you for your suggestion. We have now conducted rheological test and contact angle test to further characterize the materials. According to the rheological test, addition of the crosslinking points to SHEs were observed to enhance the mechanical strength of the polymer at both 25°C and 37°C . It further implied the mechanical adjustability of SHEs. To assess the response of the material to aqueous environments, water contact angle measurements were performed. The contact angles (92° – 99°) revealed the hydrophobicity of the materials. We also used SEM to characterize the material. From the images of SHE (using SHE2) before subcutaneous degradation experiment (Pre-implantation) and on day 35 of subcutaneous degradation experiment (Post-implantation), micro-holes appeared on the surface of SHE after 35-day implantation in vivo, indicating the degradation behavior of materials. The above results have been added to the text (manuscript-page10-line10).

5. The crosslinking structure can reduce the flexibility of polymer chain, thereby influence the self-healing performance. There should be a tradeoff between mechanical strength and self-healing property. How to ensure both the healing efficiency and toughness of this elastomer? What's the healing efficiency of three crosslinking degrees of elastomers? More explanations should be added.

Response:

Thank you for your question and suggestion. Indeed, the contradiction between the self-healing and mechanical properties of materials is one of the important issues in the development of self-healing materials. Here, PTMEG is selected as the soft segment because its flexible chains; IPDI is selected as the hard segment because its bulky structure inhibits the crystallization and increase the chain mobility, as well as derived urethane has relatively high dynamic due to the steric influence of the cyclohexyl ring. All these factors can improve self-healing. More importantly, a large number of dimethylglyoxime–urethane groups exist in the polymer backbone rather than the cross-linking points, so that the networks can be reorganized more easily. As a result, the self-healing efficiency of SHE0.2, SHE0.5 and SHE1 are over 80% ($89\% \pm 16\%$ for SHE0.2, $85\% \pm 9\%$ for SHE0.5, $94\% \pm 5\%$ for SHE1). When the degree of cross-linking increases to a higher level, the T_g (20.6°C) of SHE2 is close to the ambient temperature, which makes the chain movement difficult, resulting in lower self-healing efficiency ($37\% \pm 5\%$ for SHE2).

6. DBTDL (dibutyltin dilaurate) catalyst is toxic. Is DBTDL thoroughly removed after the reaction? The release of DBTDL should be considered during degradation.

Response:

Thank you for your question. SHE0, SHE0.2, SHE0.5, SHE1 and SHE2 in the text are synthesized under the condition of no catalyst, and the synthesis of c-SHE uses catalyst (DBTDL) to illustrate the potential harm of the catalyst to the living body.

3. Original reviewer #3 comments (remarks to the author):

Summary:

In this manuscript, Jiang and coauthors present applications of self healing elastomers in a variety of models of clinical conditions. They describe the synthesis of the polymers and their characterization, and they report biocompatibility data.

The authors performed an acute sciatic nerve cut and repair using either 10-0 sutures or with a single 10-0 suture plus SHE0.2, and included a sham control as well as an unreconstructed defect control. They assessed walking tracks and SFI, as well as neurofilament staining. They report shorter suture time, and improved SFI at six weeks postop.

Comments:

As my expertise lies in nerve regeneration I will confine my comments to those involving the nerve components of the work.

Overall I found the work to be novel and interesting.

1. I understand that the authors oriented the sciatic nerve in the SHE0.2 groups to prevent rotational misalignment, but it is not stated whether this was also done in the suture group and this should be clarified. If the nerve coaptations in the suture group were anatomically misaligned then that would account for many of the differences reported in the outcome metrics.

2. I believe the claim that the technique is faster than suturing, but I would add that the magnitude of this difference is small in terms of actual minutes.

3. The authors do not show the time course of the SFI data (such as preop, short term following surgery, and other time points).

4. The n is not stated.

5. Panel K and L show greater NF200 expression, but histomorphometry (apart from axon density, which is derived and not measured directly) is not reported, nor is retrograde labeling of neurons that regenerated their axons. It is not clear how many biological replicates were used for each metric, as the *n* varies between panels.
6. Given the popularity of fibrin glue polymers in nerve surgery I would like to see a comparison between cut and repair with fibrin glue vs. SHE, since that would represent the main source of competition in this clinical space (not sutures, necessarily). I suspect that there would be no difference in metrics of nerve regeneration in such an experiment.

Response to reviewer 3#

Comments:

As my expertise lies in nerve regeneration I will confine my comments to those involving the nerve components of the work. Overall I found the work to be novel and interesting.

Response:

Thank you for your professional comments on peripheral nerve regeneration which were great help for our improvement and future researches. The response details were demonstrated in following part.

Questions and response:

1. I understand that the authors oriented the sciatic nerve in the SHE0.2 groups to prevent rotational misalignment, but it is not stated whether this was also done in the suture group and this should be clarified. If the nerve coaptations in the suture group were anatomically misaligned then that would account for many of the differences - reported in the outcome metrics.

Response:

Thank you for your suggestion. The sciatic nerve anastomosis by suturing in this study was performed by following the instruction of surgery teaching video - *Microsurgical Repair of the Rat Sciatic Nerve*- which was produced by the *microsurgery laboratory, department of orthopedic surgery, Columbia-Presbyterian medical center*. Briefly, for avoiding distortion between nerve stumps, the nerve anastomosis auxiliary fixator was applied and the first stitch between nerve stumps was functioned as a guide line. Besides, the unique position of the capillary on the surface of the nerve epineurium could be deemed as a compass for suturing anastomosis.

2. I believe the claim that the technique is faster than suturing, but I would add that the magnitude of this difference is small in terms of actual minutes.

Response:

Thank you for your suggestion. We agree with your point and add that the magnitude of difference is small in terms of minutes in discussion part of revised manuscript (manuscript-page17-line22). In this study, the time of neuroanastomosis by SHE was reduced by 3/4 (from 12min in suture group to 3min in SHE group) compared to the neuroanastomosis by suture. It really seems that the magnitude of time difference is

small in scale of 9 minutes for single peripheral nerve anastomosis. While, perhaps, we can consider some special clinical situation such as multiple peripheral nerve (including median nerve, ulnar nerve, radial nerve and femoral nerve) damage in case of extensive limb injury or even limb mutilation. Simultaneous anastomosis of multiple peripheral nerves will take plenty of time from patient and surgeon. The scale of time saving from SHE may seem significant in that situation.

3. The authors do not show the time course of the SFI data (such as preop, short term -following surgery, and other time points).

Response:

Thank you for your suggestion. To present more specific SFI improve progression, additional time courses of the SFI were added in groups, including time of pre-operation, 2 weeks post of surgery, 4 weeks post of surgery and 6 weeks post of surgery. The analysis of nerve recovery SFI showed exciting results of SFI improvement in SHE group within 6 weeks post of surgery. Compared to other four group, the SFI of SHE group improved over time significantly. While the recovery of SFI in groups of suture, fibrin glue and conduit was proceeding slowly. The details of added SFI data were presented in the response and supplement Figure S17.

Time course of SFI in five groups

4. The n is not stated.

Response:

The **n** of each group was illustrated in Figure legends of revised manuscript and supplement figures.

5. Panel K and L show greater NF200 expression, but histomorphometry (apart from axon density, which is derived and not measured directly) is not reported, nor is retrograde labeling of neurons that regenerated their axons.

Response:

Thank you for your suggestion. The specific histomorphometry analysis were added in supplement part, including ultrastructure imaging of transmission electron microscope (TEM) for quantitative and qualitative analysis of axons. Besides, retrograde labeling was performed by using *Fluorogold* reagent based on previous studies¹⁻³. The results of TEM showed that myelin thickness was significantly higher in the SHE group than in the suture, fibrin-gluce and PCL-conduit group. The diameter of myelinated axons was larger in the SHE group than in the suture, fibrin-gluce and PCL-conduit group (Fig.S24). Retrograde labeling by fluorogold showed that the

Fluoro-gold (FG) retrograde tracing in SHE, fibrin-gluce, suture and PCL-conduit

number of motor neurons in the spinal anterior horns (SAHs) was higher in the SHE0.2 group than in the suture, fibrin-glue and PCL-conduit groups. Besides, the number of DRG neurons in the SHE0.2 group was higher than that in the suture, fibrin-glue and PCL-conduit groups (Fig.S23). The detail of results was presented in manuscript-page13-line28.

Sciatic nerve photomicrographs of transmission electron microscope (TEM) in sham, defect, suture, SHE, fibrin-glue and PCL-conduit group.

6. Given the popularity of fibrin glue polymers in nerve surgery I would like to see a comparison between cut and repair with fibrin glue vs. SHE, since that would represent the main source of competition in this clinical space (not sutures, necessarily). I suspect that there would be no difference in metrics of nerve regeneration in such an experiment.

Response:

Thank you for your suggestion. To further illustrate practicability of SHE in peripheral neuroanastomosis, we compared the nerve regeneration effect between SHE and fibrin-glue. We established the nerve amputation model and performed the neuroanastomosis with fibrin-glue followed by relieving tension between two stumps. Specifically, after amputation of sciatic nerve, we relieved the tension between stumps by same method as in SHE group, then we aligned the stumps of nerve by avoiding distortion and injected 0.5ml fibrin-glue around anastomotic area based on previous reports^{4,5}. The analysis of nerve recovery SFI showed that the efficiency of fibrin-glue for nerve recovery was inferior to SHE. While the results of nerve recovery in fibrin-glue was no difference when compared with suture, which was similar to previous report⁴. Sufficient mechanic provided by SHE can resisted against to the extrusion or contraction induced by surrounding limb muscle, while sole gel-like fibrin-glue hardly provided mechanic support as SHE did⁶, which might explain the better result of nerve function in SHE than fibrin-glue. The details of result and discussion were presented at manuscript page13-line6, page13-line11, page13-line18, page13-line25, page14-line5, page18-line6, page18-line22.

1. Wang, S., *et al.* Donor nerve axotomy and axonal regeneration after end-to-side neurorrhaphy in a rodent model. *J Neurosurg* **130**, 197-206 (2018).
2. Haastert, K., Joswig, H., Jäschke, K.A., Samii, M. & Grothe, C. Nerve repair by end-to-side nerve coaptation: histologic and morphometric evaluation of axonal origin in a rat sciatic nerve model. *Neurosurgery* **66**, 567-576; discussion 576-567 (2010).
3. Yang, Y., *et al.* Development and evaluation of silk fibroin-based nerve grafts used for peripheral nerve regeneration. *Biomaterials* **28**, 5526-5535 (2007).
4. Menovsky, T. & Beek, J.F. Laser, fibrin glue, or suture repair of peripheral nerves: a comparative functional, histological, and morphometric study in the rat sciatic nerve. *J Neurosurg* **95**, 694-699 (2001).
5. Koulaxouzidis, G., Reim, G. & Witzel, C. Fibrin glue repair leads to enhanced axonal elongation during early peripheral nerve regeneration in an in vivo mouse model. *Neural Regen Res* **10**, 1166-1171 (2015).
6. Topp, K.S. & Boyd, B.S. Structure and biomechanics of peripheral nerves: nerve responses to physical stresses and implications for physical therapist practice. *Phys Ther* **86**, 92-109 (2006).

4. Original reviewer #4 comments (remarks to the author):

First the authors should be congratulated for putting together a very broad clinically oriented set of in vivo experiments examining the self healing elastomers they have developed.

From my review I have three concerns.

1. Grammar and syntax are poor and in some cases make it difficult to follow the logic of the discussion.

*2. The self healing elastomers they describe are not novel, and there is an implication that they are. They were well described in *Advanced Materials* in 2019 for instance.*

A Highly Efficient Self-Healing Elastomer with Unprecedented Mechanical Properties <https://doi.org/10.1002/adma.201901402> describes the very same chemistry that has been presented.

3. I believe an important study group is missing. Many biomaterials are conductive for tissue and a control group of a non 'self healing biomaterial' should have been used. For instance, it is well known and studied that polymeric nerve conduits can have beneficial effects on nerve conduction and healing. Why were they not used as controls? Without that group it is not clear what the real benefit of these polymeric materials are.

Response to reviewer 4#

Comments:

First the authors should be congratulated for putting together a very broad clinically oriented set of in vivo experiments examining the self healing elastomers they have developed.

Response:

Thank you for your professional comments for improving the manuscript. The response details were demonstrated in following part.

Questions and response:

1. Grammar and syntax are poor and in some cases make it difficult to follow the logic of the discussion.

Response:

Thank you for your suggestion. The errors of grammar and syntax in the draft were checked and corrected by author and co-authors before sending to *Nature Research* for native language editing. The logic of the discussion was re-organized, which make sure that readers can easily follow.

2. The self healing elastomers they describe are not novel, and there is an implication that they are. They were well described in *Advanced Materials* in 2019 for instance. A Highly Efficient Self-Healing Elastomer with Unprecedented Mechanical Properties <https://doi.org/10.1002/adma.201901402> describes the very same chemistry that has been presented.

Response:

Thank you for your question. The self-healing elastomers of this study were different to that of our previous study(*A Highly Efficient Self-Healing Elastomer with Unprecedented Mechanical Properties*)¹. In the previous study, we utilized catalyst (dibutyltin dilaurate) to fabricate self-healing elastomer. However, for applications *in vivo*, the toxic catalyst used in previous study was seemed inappropriate. Therefore, we improved the synthesis protocol (free of catalyst) and fabricated a biofriendly self-healing elastomer. Hence, from the point of gradients in synthesis, the self-healing elastomers in this study were different to the previous study even characterized with same chemistry.

3. I believe an important study group is missing. Many biomaterials are conductive for tissue and a control group of a non 'self healing biomaterial' should have been used. For instance, it is well known and studied that polymeric nerve conduits can have beneficial effects on nerve conduction and healing. Why were they not used as controls? Without that group it is not clear what the real benefit of these polymeric materials are.

Response:

Thank you for your suggestion. To strengthen the viewpoint of SHE practicability in peripheral neuroanastomosis, we compared the nerve healing effect between SHE and traditional polymeric nerve conduit. Based on previous studies²⁻⁴, we used PCL, which is approved in medical apparatus fabrication by FDA, as main component of

nerve conduit. We established the nerve amputation model and performed the neuroanastomosis with PCL-conduit followed by relieving tension between two stumps. The operation of nerve conduit implantation was referred to previous studies². Specifically, after amputation of sciatic nerve, we relieved the tension between stumps by same method as in SHE group, then we inserted the two nerve stumps into conduit gently and without distortion before fixing the conduit with nerve by suture. The analysis of nerve recovery showed inferior results in PCL-conduit group when compared with SHE group. Lower moduli of SHE than PCL-conduit reduced the mechanical stimuli to sciatic nerve, providing favorable mechanical matching and conduit structure for nerve ends protection away from tissue invasion^{5,6}, which might explain the better result of nerve function in SHE rather than PCL-conduit group. The details of result and discussion were presented at manuscript page13-line6, page13-line11, page13-line18, page13-line25, page14-line5, page18-line6, page18-line19.

1. Zhang, L., *et al.* A Highly Efficient Self-Healing Elastomer with Unprecedented Mechanical Properties. *Advanced Materials* **31**(2019).
2. Qian, Y., *et al.* An integrated multi-layer 3D-fabrication of PDA/RGD coated graphene loaded PCL nanoscaffold for peripheral nerve restoration. *Nature Communications* **9**, 323 (2018).
3. Lee, B.K., *et al.* End-to-side neurorrhaphy using an electrospun PCL/collagen nerve conduit for complex peripheral motor nerve regeneration. *Biomaterials* **33**, 9027-9036 (2012).
4. Fadia, N.B., *et al.* Long-gap peripheral nerve repair through sustained release of a neurotrophic factor in nonhuman primates. *Sci Transl Med* **12**(2020).
5. Alderton, G. Mechanical tissue matching. *Science* **366**, 967 (2019).
6. Wang, L., *et al.* Functionalized helical fibre bundles of carbon nanotubes as electrochemical sensors for long-term in vivo monitoring of multiple disease biomarkers. *Nat Biomed Eng* **4**, 159-171 (2020).

Best regards,

Xiaofeng Ye, M.D& Ph.D.
Dept. Cardiac Surgery, Ruijin Hospital
Shanghai Jiao Tong University, School of Medicine
No 197 Ruijin Er Rd.

Shanghai 200025, China

REVIEWER COMMENTS

Reviewer #2 (Remarks to the Author):

The authors have properly revised the manuscript. Now it should be publishable.

Reviewer #3 (Remarks to the Author):

Comments for author:

In the revised version, the authors have provided extended sciatic function index (SFI), histomorphometry and retrograde labelling data.

1. SFI: These data show that SFI improves dramatically over a short period of time compared with the suture group and others. If true, this represents a significant advance in the field of nerve regeneration. I am not sure how to explain this gain in the rate of recovery, nor is an explanation provided as to how this improvement may be possible.

2. Histomorphometry: The revised data show a marked improvement of the myelin thickness, again within a short period of time. I am not sure how this is being promoted, but this is a noteworthy finding that should be probed from a molecular standpoint.

3. Retrograde labeling: The findings with fluorogold indicate improved recovery of both sensory and motor neuron regeneration. The sampling method used by the authors is not completely described. The data from FastBlue was not visible to me, either in the revised manuscript or in the Supplement.

4. Minor: the term "anastomosis" is outmoded for nerve surgery and should be updated to "coapatation."

Reviewer #6 (Remarks to the Author):

The manuscript presents interesting data with respect to the beneficial biocompatibility of their elastomers, but the comments of reviewer #4 related to the language and added value of using self-healing polymers were not addressed adequately. Despite the language correction, it is still too evident that the manuscript has not been written by native speakers.

To provide convincing evidence for the added value of self-healing elastomer vs. on-selfhealing control, the authors used PCL conduits as non-self-healing controls. Although the regenerative performance of the self-healing elastomers was improved relative to the non-self-healing PCL control, the authors attribute this difference in performance to the lower elastic modulus of their self-healing elastomers without referring to differences in self-healing capacity. The afore-mentioned difference in elastic modulus between PCL and SHE was not quantified as well. Overall, the discussion on the differences in regenerative capacity between SHE and controls such as PCL remains somewhat superficial without deciphering fundamental mechanisms underlying tissue regeneration.

Consequently, the added value of the self-healing capacity of SHE for tissue regeneration remains elusive. Since the authors present this self-healing capacity of their elastomers as one of the main novelties of their work, the main claim of the authors is not yet substantiated by solid data highlighting the beneficial effect of self-healing polymers for tissue regeneration.

Reviewer #7 (Remarks to the Author):

The authors have done well to answer many of the reviewer concerns related to the aneurysm models. However, some of the results warrant further clarification.

1. For the MRI measurements, how were the maximum diameters derived? How long of a segment in each animal was evaluated?
2. For the morphology measures, what precautions were taken to ensure the fixative solution was injected with a fixed, physiological pressure in order to mimic in vivo diameters as well as enable comparison between samples.
3. Quantification of immunohistochemistry is challenging, especially for a single layer of cells like the endothelium. How was eNOS staining quantified and in how many sections was this performed? A second quantitative measure such as Western blot or even RT-PCR would engender more confidence in the claim that eNOS levels are altered.
4. The elastin grading system needs further explanation. How was this used to quantify elastin quantity and integrity? How many sections from the aneurysm were used for each animal? There are other examples in the literature that have quantified elastin density and breaks from similar images.
5. The authors chose a model that is not well represented in the literature compared to the elastase infusion model, the AngII model, or even the calcium chloride model. Some discussion is warranted related to how well this model relates to human disease in terms of mechanism and time course.
6. The authors miss an opportunity to better understand the mechanisms by which aortic wrapping with SHE limits aneurysm growth. External aortic wrapping has been historically documented in both the pre-clinical and clinical literature none have been satisfactorily adopted into surgical practice. Indeed, Albert Einstein famously underwent surgical wrapping of his abdominal aortic aneurysm with cellophane, only to later die of aneurysm rupture. Various materials and configurations have been tried as a method to limit aneurysm growth but most have resulted with a foreign body reaction which exacerbates the inflammation present in the native aneurysm. The model presented (as well as other murine models of AAA) is hallmarked by early inflammation which is followed by elastin breakdown and smooth muscle dropout, leading to eventual weakening of the aortic wall and vasodilatation. Given that these are all important components of human disease, it would be important to describe how the SHE affects each of these processes.
7. Angiogenic processes have been shown to enhance inflammation and accelerate AAA in similar experimental models as well as benefit demonstrated with anti-angiogenic agents. What effects does SHE have on medial and peri-adventitial neovascularization?

We have carefully studied these comments which are all valuable and very helpful for revising and improving our research. Following the professional comments and suggestions from editors and reviewers, we have made several corrections and added more specific contents in revised manuscript. The content alterations were marked with color of **RED** and grammar corrections were marked with color of **GREEN** in the manuscript and the responses for comments of reviewers were listed orderly in following part of this letter.

I. Summary of main added experiment contents

Part	Added content	Purpose	Response to
Material	Elastic modulus of PCL and silicone rubber	To summarize the elastic modulus of materials used in this study	Reviewer #6
Aneurysm	Inflammation, apoptosis and neovascularization	To clarify the mechanism of SHE wrapping effect	Reviewer #7
	Aneurysm wrapping by non-self-healing material (silicone rubber)	To present beneficial effect of self-healing elastomer when compared to non-self-healing material	Reviewer #6
Nerve coaptation	Apoptosis of Schwann cell (dual labeling of Schwann cell and caspase-3, qRT-PCR of caspase-3 transcription level)	To illustrate the underlying mechanism of nerve regeneration by self-healing elastomer	Reviewer #3 and #6
	LuxolFastBlue staining of axons	To complete the data of histomorphometry of axons	Reviewer #3

II. Details of response

Reviewer #2 (Remarks to the Author):

The authors have properly revised the manuscript. Now it should be publishable.

Response:

Thank you for your work.

Reviewer #3 (Remarks to the Author)

Comments for author:

In the revised version, the authors have provided extended sciatic function index (SFI), histomorphometry and retrograde labelling data.

1. SFI: These data show that SFI improves dramatically over a short period of time compared with the suture group and others. If true, this represents a significant advance in the field of nerve regeneration. I am not sure how to explain this gain in the rate of recovery, nor is an explanation provided as to how this improvement may be possible.

Response:

Thank for your suggestions. When compared to the long-defect peripheral nerve regeneration in other studies¹⁻³, our data show a satisfied improvement in a short period. However, we should admit that it is still difficult to restore the nerves to normal function after the severed injury in a short time. But what makes us gratified is that while achieving a similar nerve recovery efficiency compared to other recorded literatures⁴⁻⁷, we do provide a simpler and easier nerve coaptation method which is worth promotion and further translation.

The satisfied regeneration of peripheral nerves after injury mainly depends on the regeneration microenvironment in which the nerves grow. The discussion about why SHE can improve the recovery of nerve mainly lie in following respects and summarized in revised manuscript (Page19, line34):

First, the conduit formed by SHE essentially forms a nerve regeneration chamber. The chamber is a hollow lumen that protects and supports the growth of nerves. After nerve injury, the exudate secreted by stumped nerve end containing factors including neurite growth factor and neuron trophic factor can promote nerve regeneration. The existence of chamber can maintain a high concentration of these factors, improving proximal axons selective regeneration and avoiding the misconnection between sensory and motor axons⁸.

Second, the good mechanical strength of SHE can resist the compression of tissues and prevent the intrusion of surrounding scars, protecting regenerated fibers and

providing a good micro-environment for nerve regeneration without external interference^{9,10}.

Third, as the most exciting highlight of this part of research, compared to other non-self-healing materials, the SHE can take advantage of its excellent self-healing property to shape in real time according to the shape of the anastomosed nerve, which can be deemed as customization in other word. The material and tissue can achieve the most perfect physical shape fit without any adverse factors such as chronic compression around the nerve, because the diameter of the SHE conduit naturally matches the target nerve stump size.

Fourth, with the help of SHE, the “LEGO-Playing” nerve coaptation method does not require surgical suture, which avoid extra surgical trauma on neve and prevent the possibility of ineffective nerve growth or neuroma formation caused by forced docking of suture.

Last but not least, we observed a significant decrease of the quality of apoptotic Schwann cell and mRNA transcription levels of apoptotic protein caspase-3 in SHE group by the dual labeling (S100 β and caspase-3) immunofluorescence staining and quantitative real-time PCR of caspase-3. The details can be seen in second “*Question and Answer*” in following content. It can be deduced that the “LEGO-playing” nerve coaptation method can improve nerve recovery after severed injury by alleviating Schwann-cells apoptosis.

In a summary, by seamless LEGO-playing coaptation protocol, the self-healing elastomer provided a nerve regeneration chamber and material-tissue-matched mechanical protection, reducing the Schwann cell apoptosis in the process of nerve regeneration, and improving the recovery of peripheral nerves after severed injury.

2. Histomorphometry: The revised data show a marked improvement of the myelin thickness, again within a short period of time. I am not sure how this is being promoted, but this is a noteworthy finding that should be probed from a molecular standpoint.

Response:

Thank you for your suggestion. To uncover the underlying mechanism of peripheral nerve improvement after coaptation by SHE, we performed dual labeling (S100 β and caspase-3) immunofluorescence staining and quantitative real-time PCR of caspase-3 to analyze the apoptosis of Schwann cell in each group and we observed apoptosis difference among groups.

Schwann cells are the most important cells for repairing peripheral nerves after injury^{11,12}. The mechanical stimulation received during nerve repair usually affects the proliferation and apoptosis of Schwann cells¹³⁻¹⁵. We observed differences in apoptosis among the groups of neural Schwann cells, including differences in the

transcription and expression levels of Schwann cell apoptotic protein caspase-3. Specifically (shown in following figure), we observed that less apoptotic Schwann-cell appeared in the SHE-group under fluorescence microscope, and the caspase-3 mRNA transcription level was also the lowest in the SHE-group than others. While the apoptosis levels including apoptotic Schwann cell number and transcription level of caspase-3 were increased significantly in PCL conduit group.

Based on the theory of the previous studies^{13,14}, effective mechanical protection in SHE-group rather than high mechanical stimuli in conduit group or absence of mechanical protection in others can alleviate the level of apoptosis and improve nerve recovery. In a summary, based on the self-healing elastomer, the “LEGO-playing” nerve coaptation method can improve nerve recovery after severed injury by alleviating Schwann-cells apoptosis.

Supplementary Figure 31. Apoptosis of Schwann cell and mRNA transcription of caspase-3

(A) Immunofluorescence staining of S100 and Caspase-3 in SHE, glue, suture and conduit groups.

Green represents S100 positive Schwann cell and red represents caspase-3 positive cell. White arrows illustrate the typical caspase-3 positive Schwann cells which are mixed with both green and red. Scale bar = 100 μ m.

(B) Statistic histogram of counting number of apoptotic Schwann cell based on images of immunofluorescence staining in sham, SHE, glue, suture and conduit groups.

(C) Statistic histogram of mRNA relative expression foldchange of caspase-3 based on results of quantitative real-time PCR in sham, SHE, glue, suture and conduit groups. GAPDH was used as internal reference. (n = 5 in each group).

* $p < 0.05$, ns = no significance.

3. Retrograde labeling: The findings with fluorogold indicate improved recovery of both sensory and motor neuron regeneration. The sampling method used by the authors is not completely described. The data from FastBlue was not visible to me, either in the revised manuscript or in the Supplement.

Response:

1. Retrograde labeling sampling method: according to the previous study¹⁶ and product manual, the powder of Fluoro-Gold was dissolved in 0.9% saline. For retrograde labeling of ganglia and spinal cord connected with sciatic nerve, the distal sciatic nerve was cut and dipped in aq. 5% solution of fluoro-gold. The survival time of animals was 4 days post of labeling. At the day 4 post of labeling, the animal was euthanized and then fixed with paraformaldehyde. We exposed the whole course of the sciatic nerve under the microscope and found the ganglion corresponding to the sciatic nerve. Meanwhile, with the help of rongeurs, we opened the lamina of the animal's spine to expose the spinal cord. The ganglions and spinal cord from L4-L6 were extracted by microscissor. The specimens of ganglions and spinal cord were frozen and sectioned after dehydration with sucrose in a dark room. The slices were visualized under the fluorescence microscope using a wide band ultraviolet (UV) excitation filter and positive neurons emitted yellow light when excited by UV light. The detail of retrograde labeling method was added in the supplementary method.

2. The data of LuxolFastBlue were summarized in supplement file (Supplementary Fig. 29A) and displayed in following content

4. Minor: the term “anastomosis” is outmoded for nerve surgery and should be updated to “coaptation.”

Response:

Thank you for your suggestion. The term of “anastomosis” has been replaced by “coaptation” in the revised manuscript.

1. Lee, B.K., *et al.* End-to-side neurorrhaphy using an electrospun PCL/collagen nerve conduit for complex peripheral motor nerve regeneration. *Biomaterials* **33**, 9027-9036 (2012).
2. Zhang, W., *et al.* Repairing sciatic nerve injury with an EPO-loaded nerve conduit and sandwiched-in strategy of transplanting mesenchymal stem cells. *Biomaterials* **142**, 90-100 (2017).
3. Xie, H., *et al.* A silk sericin/silicone nerve guidance conduit promotes regeneration of a transected sciatic nerve. *Adv Healthc Mater* **4**, 2195-2205 (2015).
4. Lin, Y.F., *et al.* Effect of exogenous spastin combined with polyethylene glycol on sciatic nerve injury. *Neural Regen Res* **14**, 1271-1279 (2019).
5. Cheng, X.L., *et al.* The longitudinal epineural incision and complete nerve transection method for modeling sciatic nerve injury. *Neural Regen Res* **10**, 1663-1668 (2015).
6. Wang, P., Zhang, Y., Zhao, J. & Jiang, B. Intramuscular injection of bone marrow mesenchymal stem cells with small gap neurorrhaphy for peripheral nerve repair. *Neurosci Lett* **585**, 119-125 (2015).
7. Qian, Y., *et al.* An integrated multi-layer 3D-fabrication of PDA/RGD coated graphene loaded PCL nanoscaffold for peripheral nerve restoration. *Nature Communications* **9**, 323 (2018).
8. Lundborg, G. Nerve regeneration and repair. A review. *Acta Orthop Scand* **58**, 145-169 (1987).
9. Evans, G.R. Approaches to tissue engineered peripheral nerve. *Clin Plast Surg* **30**, 559-563, viii (2003).
10. Belkas, J.S., Munro, C.A., Shoichet, M.S., Johnston, M. & Midha, R. Long-term in vivo biomechanical properties and biocompatibility of poly(2-hydroxyethyl methacrylate-co-methyl methacrylate) nerve conduits. *Biomaterials* **26**, 1741-1749 (2005).
11. Zhao, Z., Li, X. & Li, Q. Curcumin accelerates the repair of sciatic nerve injury in rats through reducing Schwann cells apoptosis and promoting myelination. *Biomed Pharmacother* **92**, 1103-1110 (2017).
12. Butts, B.D., Houde, C. & Mehmet, H. Maturation-dependent sensitivity of oligodendrocyte lineage cells to apoptosis: implications for normal development and disease. *Cell Death Differ* **15**, 1178-1186 (2008).
13. Gupta, R. & Steward, O. Chronic nerve compression induces concurrent apoptosis and proliferation of Schwann cells. *J Comp Neurol* **461**, 174-186 (2003).
14. Pham, K. & Gupta, R. Understanding the mechanisms of entrapment neuropathies. Review article. *Neurosurg Focus* **26**, E7 (2009).
15. Lin, M.Y., *et al.* Biophysical stimulation induces demyelination via an integrin-dependent mechanism. *Ann Neurol* **72**, 112-123 (2012).
16. Guo, D., *et al.* Therapeutic Effect of Vinorine on Sciatic Nerve Injured Rat. *Neurochem Res* **43**, 375-386 (2018).

Reviewer #6 (Remarks to the Author)

1. The manuscript presents interesting data with respect to the beneficial biocompatibility of their elastomers, but the comments of reviewer #4 related to the language and added value of using self-healing polymers were not addressed adequately. Despite the language correction, it is still too evident that the manuscript has not been written by native speakers.

Response:

Thank you for your suggestion. We have corrected the grammar and collocation errors in the revised manuscript. In addition, the revised manuscript has been language polished by gold-language editing from *nature research editing service* before re-submission.

SPRINGER NATURE
Author Services Editing Certificate

This document certifies that the manuscript

The Self-healing Polyurethane-elastomer with Mechanical Tunability for Multiple Biomedical Applications in vivo

prepared by the authors

Chenyu Jiang, Luzhi Zhang, Qi Yang, Shixing Huang, Hongpeng Shi, Qiang Long,...

was edited for proper English language, grammar, punctuation, spelling, and overall style by one or more of the highly qualified native English speaking editors at SNAS.

This certificate was issued on **March 16, 2021** and may be verified on the SNAS website using the verification code **4873-F203-D6F7-CDAB-B99P**.

Neither the research content nor the authors' intentions were altered in any way during the editing process. Documents receiving this certification should be English-ready for publication; however, the author has the ability to accept or reject our suggestions and changes. To verify the final SNAS edited version, please visit our verification page at secure.authorservices.springernature.com/certificate/verify.
If you have any questions or concerns about this edited document, please contact SNAS at support@as.springernature.com.

SNAS provides a range of editing, translation, and manuscript services for researchers and publishers around the world. For more information about our company, services, and partner discounts, please visit authorservices.springernature.com.

2. To provide convincing evidence for the added value of self-healing elastomer vs. non-self-healing control, the authors used PCL conduits as non-self-healing controls. Although the regenerative performance of the self-healing elastomers was improved relative to the non-self-healing PCL control, the authors attribute this difference in performance to the lower elastic modulus of their self-healing elastomers without referring to differences in self-healing capacity. The afore-mentioned difference in elastic modulus between PCL and SHE was not quantified as well. Overall, the discussion on the differences in regenerative capacity between SHE and controls such as PCL remains somewhat superficial without deciphering fundamental mechanisms underlying tissue regeneration.

Consequently, the added value of the self-healing capacity of SHE for tissue regeneration remains elusive. Since the authors present this self-healing capacity of their elastomers as one of the main novelties of their work, the main claim of the authors is not yet substantiated by solid data highlighting the beneficial effect of self-healing polymers for tissue regeneration.

Response:

1. Thank you for your suggestion. The results of the mechanical differences between PCL and SHE have been summarized in the supplement file.

Table1. Elastic Modulus Summary of materials used in this study

Material	Elastic Modulus	Application in vivo
SHE 0	172 ± 61 kPa	—
SHE 0.2	612 ± 199 kPa	Peripheral nerve coaptation
SHE 0.5	1166 ± 198 kPa	Aneurysm wrapping
SHE 1	1516 ± 227 kPa	
SHE 2	3724 ± 787 kPa	Sternum immobilization
Silicone rubber	2010 ± 291 kPa	As non-self-healing material control in aneurysm wrapping
PCL	154 ± 13 MPa	As non-self-healing nerve conduit control in peripheral nerve coaptation

2. To uncover the underlying mechanism of peripheral nerve improvement after coaptation by SHE, we performed dual labeling (S100 β and caspase-3) immunofluorescence staining and quantitative real-time PCR of caspase-3 to analyze the apoptosis of Schwann cell in each group and we observed apoptosis difference among groups. The detail results can be found in revised manuscript (Page 14, line 15; Supplementary Fig. 31)

Schwann cells are the most important cells for repairing peripheral nerves after injury¹. The mechanical stimulation received during nerve repair usually affects the proliferation and apoptosis of Schwann cells^{2,3}. Hence, to explain the previous experiment's phenomenon of different nerve recovery rate among the groups, we performed the dual labeling (S100 β and caspase-3) immunofluorescence staining and quantitative real-time PCR of caspase-3 to analyze the apoptosis in each group. We observed differences in apoptosis among the groups of Schwann cells, including differences in the number of apoptotic Schwann cell and the transcription levels of apoptotic protein caspase-3. We observed that less apoptotic Schwann-cell appeared in the SHE-group under fluorescence microscope, and the caspase-3 mRNA transcription level was also the lowest in the SHE group than others. While the apoptosis levels including apoptotic Schwann cell number and transcription level of caspase-3 were increased significantly in PCL conduit group.

Based on the theory of the previous studies^{3,4}, effective mechanical protection in SHE -group rather than high mechanical stimuli in conduit group or absence of mechanical protection in suture and glue groups can alleviate the level of apoptosis and be benefit to improve nerve recovery. In a summary, based on the self-healing elastomer, the “LEGO-playing” nerve coaptation method can improve nerve recovery after severed injury by alleviating Schwann-cells apoptosis.

Supplementary Figure 31. Apoptosis of Schwann cell and mRNA transcription of caspase-3

(A) Immunofluorescence staining of S100 and caspase-3 in SHE, glue, suture and conduit groups. Green represents S100 positive Schwann cell and red represents caspase-3 positive cell. White arrows illustrate the typical caspase-3 positive Schwann cells which are mixed with both green and red. Scale bar = 100 μ m. (B) Statistic histogram of counting number of apoptotic Schwann cell based on images of immunofluorescence staining in sham, SHE, glue, suture and conduit groups. (C) Statistic histogram of mRNA relative expression foldchange of caspase-3 based on results of quantitative real-time PCR in sham, SHE, glue, suture and conduit groups. GAPDH was used as internal reference. (n = 5 in each group). * p < 0.05, ns = no significance.

3. In this study, we mainly use the self-healing ability of materials to solve or improve the current therapies of surgical diseases. The original intention of using self-healing capacity is to achieve rapid shaping and integrating of materials without surgical suture or glue bonding during surgery. For example, the size and shape of target tissue or organs such as abdominal aneurysm and peripheral nerve cannot be predicted precisely before surgery, while we can make fine-tuning on size or shape of implants to match the target tissues by rapid self-healing ability.

Compared to non-self-healing materials that need to be prefabricated into the particular shape and size before surgery, the self-healing ability of SHE allows the materials to be real-time shaped and integrated on the basis of original physical size of target tissue during surgery. Specifically, in this study, the rapid self-healing property actualizes the LEGO-playing coaptation which cannot be replaced by other non-self-healing materials, greatly simplifying the procedure of peripheral nerve coaptation, reducing the time of operation, avoiding chronic nerve compression and shortening the learning curve of young neurosurgeons.

Moreover, the self-healed material can be deemed as an intact one from macroscopic physics to inner chemical covalent bonds after self-healing process. That is still quite different from non-self-healing materials which are only stitched together to achieve physical splicing rather than macro- and micro-structure integrity. As the results of the added experiment (used silicone rubber as non-self-healing control material in aneurysm wrapping), the effect against aneurysm progress was slightly superior in SHEs than silicone rubber even with similar mechanical strength.

In a summary, the rapid self-healing ability would realize real-time material shaping and bring about method changes of the surgery, showing the potential value of application in surgery filed.

1. Butts, B.D., Houde, C. & Mehmet, H. Maturation-dependent sensitivity of oligodendrocyte lineage cells to apoptosis: implications for normal development and disease. *Cell Death Differ* **15**, 1178-1186 (2008).
2. Lin, M.Y., *et al.* Biophysical stimulation induces demyelination via an integrin-dependent mechanism. *Ann Neurol* **72**, 112-123 (2012).
3. Gupta, R. & Steward, O. Chronic nerve compression induces concurrent apoptosis and proliferation of Schwann cells. *J Comp Neurol* **461**, 174-186 (2003).
4. Pham, K. & Gupta, R. Understanding the mechanisms of entrapment neuropathies. Review article. *Neurosurg Focus* **26**, E7 (2009).

Reviewer #7 (Remarks to the Author)

The authors have done well to answer many of the reviewer concerns related to the aneurysm models. However, some of the results warrant further clarification.

1. For the MRI measurements, how were the maximum diameters derived? How long of a segment in each animal was evaluated?

Response:

Thank you for your questions. According to the previous study¹, we will determine the scan area of the abdominal aorta through the scout scanning before formal horizontal scanning. We used the sagittal image from the scout scanning to observe the shape of the abdominal aorta and determined the final positions of subsequent horizontal scanning, which ensures the image of the maximum dilation part of the abdominal aorta can be captured. The MRI scanning of each animal starts below the opening of the renal artery and ends at the bifurcation of the iliac artery. The length of the total scan segment is 15mm.

2. For the morphology measures, what precautions were taken to ensure the fixative solution was injected with a fixed, physiological pressure in order to mimic in vivo diameters as well as enable comparison between samples.

Response:

Thank you for your questions. According to published researches²⁻⁴, mice were anesthetized and left cardiac ventricles were perfused with phosphate-buffered saline followed by 4% paraformaldehyde via a syringe and catheter inserted into the left ventricle. We used a syringe (5ml) to inject liquid at a rate of 0.1ml per second. After the measurement of the electronic pressure gauge, the pressure produced by this bolus rate is about 100mmHg, which is similar to the arterial blood pressure of normal mice. These precautions can prevent blood vessels from collapsing or over-expanding. However, we still wanted to perform morphology measurement under the physiological state, hence, the measurement of aorta morphology in this study was achieved by in vivo imaging like MRI.

3. Quantification of immunohistochemistry is challenging, especially for a single layer of cells like the endothelium. How was eNOS staining quantified and in how many sections was this performed? A second quantitative measure such as Western blot or even RT-PCR would engender more confidence in the claim that eNOS levels are altered.

Response:

Thank you for your suggestions. The percentage of stained positive area of eNOS was calculated by software *ImageJ*, as follows: after importing the slice image into *ImageJ*, we followed the manual instruction of *Image J* to automatically calculate the percentage of eNOS staining positive area. The percentage value of the positive area of eNOS staining for each animal was determined by the average percentage value of 4 consecutive sections. Moreover, to precisely analyze the difference of eNOS in each group, we further performed quantitative real-time PCR to observe the mRNA of

eNOS transcription level alteration in each group. The results show that the foldchange of eNOS transcription in aneurysm was lowest among other four groups, and there is no significant foldchange difference between SHE1 and SHE0.5group.

mRNA of eNOS transcription level

GAPDH was used as internal reference. n = 5 in each group.

*** p < 0.001, ns = no significance.

4. The elastin grading system needs further explanation. How was this used to quantify elastin quantity and integrity? How many sections from the aneurysm were used for each animal? There are other examples in the literature that have quantified elastin density and breaks from similar images.

Response:

Thank you for your suggestions. According to the previous studies⁵⁻⁹, we adopted the traditional elastin grading system to evaluate elastin degradation, which is widely used in many researches. The elastin grading system is comprised of four criterions: grade1: <25% of the medial circumference digested; grade2, 25%-50% of the medial circumference digested; grade3: 50%-75% of the medial circumference digested; grade4, 75%-100% of the medial circumference digested. In order to estimate the grade of elastin degradation objectively, the image of EVG staining is equally divided into four quadrants and we grade elastin degradation based on the number of quadrants occupied by the elastin fragmentation area. An average of 4 sections per mouse was counted from each aorta examined.

5. The authors chose a model that is not well represented in the literature compared to the elastase infusion model, the AngII model, or even the calcium chloride model. Some discussion is warranted related to how well this model relates to human disease in terms of mechanism and time course.

Response:

Thank you for your suggestions. The brief discussion of aneurysm model was added in the revised manuscript (Page17, line32) and supplementary methods. The detail analysis and discussion are lied in following content.

The existing aneurysm model methods include AngII perfusion, porcine elastase method, and calcium chloride method¹⁰. However, Jonathan Golledge et al. have

concluded that each method has its shortcomings and cannot perfectly mimic the pathogenesis of human abdominal aortic aneurysms^{10,11}.

Porcine pancreatic elastase (PPE) induction is a common method to establish AAA by including intraluminal infusion and aortic adventitial application. The PPE infusion method was reported by Anidjar et al. in 1990s, which was firstly performed in Wistar rat for abdominal aortic aneurysm establishment¹². PPE infusion model involves sophisticated surgical procedure including dissecting, occluding, infusing via an arteriotomy, and artery repairing, which is far more challenging to perform than the adventitial elastase application. More importantly, it is a common phenomenon of aorta stenosis after infusion especially in mice, leading animal lower limb paralysis or even death and lowering the aneurysm model reproductivity. Applying PPE to the adventitia of the infrarenal aorta is an improved method proposed by Bhamidipati et al. for aneurysm model establishment of mice in 2012¹³. This method only needs to free the abdominal aorta and bathe it in a high concentration of elastase. Due to the simple modeling process, stable modeling effect, and high repetition rate, the adventitial elastase application has been adopted currently by increasing studies which can be found in many cardiovascular professional journals including *Circulation*^{14,15}, *Arteriosclerosis, thrombosis, and vascular biology*^{5,16} and other professional journals^{17,18}.

The aneurysm in this study was established by PPE adventitial application. It has been reported in the literature that there is no difference between PPE adventitial application and PPE infusion in the aneurysm formation time during the process of mouse aneurysm modeling¹⁹. The pathological characteristics of aneurysm established in this study were as follows: (a) the diameter of the abdominal aortic aneurysm expanded to three times than that of the original artery (Fig. 4C); (b) the middle layer of the artery was extensively destroyed (Fig. 4E); (c) a large number of inflammatory cells infiltrated (Supplementary Fig. 13A, 13B, 13E, and 13F); (d) numerous apoptosis of vascular smooth muscle cells can be observed (Supplementary Figs. 13C, 13G and 14B). The above phenomena are similar to aneurysm model reported in other studies using PPE infusion including the phenomenon of elastin degradation¹¹, macrophage infiltration¹³, and smooth muscle apoptosis²⁰, which are also the major mechanisms of human abdominal aortic aneurysm pathogenesis¹⁰.

Other modeling methods may be not suitable for the purpose of this study. The reasons are as follows: AngII model is the most common aneurysm model, which is simple and convenient without delicate surgical procedure. However, several studies suggest that angiotensin II induces an intimal tear and bleed into the aorta wall, typical of aortic dissection^{21,22}, which is not typical of human abdominal aortic aneurysm. Furthermore, angiotensin II-induced aneurysms develop in the suprarenal or thoracic aorta, while in patients, the most common site affected is the infrarenal aorta²¹. Also, the abdominal aorta above the renal artery in mice has many important organ branches and cannot be used as an experimental model for external arterial wrapping in animals. Application of calcium salts, usually CaCl₂ or calcium

phosphate, to the adventitia of the infrarenal aorta is another method of inducing AAA, but the degree of aortic diameter expansion stimulated is less marked than that caused by angiotensin II or elastase^{22,23}.

It should be admitted that the current animal models of aneurysm including AngII, elastase and calcium chloride are impossible to simulate the whole nature development of human aneurysms. Some methods induce arterial aneurysm formation involving acute inflammation of the arterial wall and the established aneurysm will appear spontaneously resolving trend after chronic phase (> 8 weeks)¹⁰.

Based on the above facts and the breadth of the research field in this research, our team comprehensively considered the advantages and disadvantages of various aneurysm models in the initial research design and finally chose a relatively stable and simple PPE adventitial application to establish aneurysm model.

6. The authors miss an opportunity to better understand the mechanisms by which aortic wrapping with SHE limits aneurysm growth. External aortic wrapping has been historically documented in both the pre-clinical and clinical literature none have been satisfactorily adopted into surgical practice. Indeed, Albert Einstein famously underwent surgical wrapping of his abdominal aortic aneurysm with cellophane, only to later die of aneurysm rupture. Various materials and configurations have been tried as a method to limit aneurysm growth but most have resulted with a foreign body reaction which exacerbates the inflammation present in the native aneurysm. The model presented (as well as other murine models of AAA) is hallmarked by early inflammation which is followed by elastin breakdown and smooth muscle dropout, leading to eventual weakening of the aortic wall and vasodilatation. Given that these are all important components of human disease, it would be important to describe how the SHE affects each of these processes.

Response:

1. Thank you for your suggestion. To better understand the underlying mechanism of SHEs wrapping against aneurysm expansion, we further analyze the effect of SHE from perspective of inflammation and smooth muscle apoptosis. The results exhibited that SHEs can prevent the further expansion of the aneurysm by alleviating the inflammation of the artery wall and reducing the apoptosis of smooth muscle cells. The detail results were summarized in revised manuscript (Page13, line7; Page18, line13) and supplement files (Supplementary Fig. 13) and also presented in following content.

2. Thank you for your comment on issue of external wrapping. The distinct pathological feature of aneurysm is the destruction of the arterial elastic layer caused by various factors, resulting in the loss of elasticity of the arterial wall. From the perspective of surgical reconstruction, external arterial wrapping is aimed to reconstruct arterial middle layer. However, current wrapping materials including

Cellophane and Dacron are characterized as high mechanical strength with absence of elastic compliance, which means too hard to artery and can cause mechanical damage to the artery like an egg striking a rock, resulting in artery wall erosion, fistulization or even rupture after long-term wrapping²⁴. The self-healing elastomer used in this study can be deemed as a bionic elastic material with similar mechanics to normal arterial wall, that is, the elastic modulus of the material is consistent with the arterial wall, and it is elastically compliant, avoiding the chronic mechanical mismatch injury.

In addition to the application of the abdominal aortic aneurysm, the external arterial wrapping can also be adopted in the other cases of arterial dilation. For example, when patients with coronary heart disease and chronic hypertension are undergoing coronary artery bypass surgery, it is usually seen that the ascending aorta dilated with disappearance of sinus tube junction which leads to aortic valve insufficiency; in order to solve this kind of regurgitation, polyester patch will be used to wrap the ascending aorta to limit its expansion and reduce or eliminate aortic regurgitation during coronary artery bypass grafting. Moreover, for patients received endovascular treatment after ascending aortic dissection, the implanted stent graft will generate a force which make the stent rebound to the greater curvature of the ascending aortic arch when stent is passing through the curved aortic arch, causing the corolla structure of the stent to become unanchored and unable to be fixed on arterial wall, at this time, the ascending aorta can be wrapped to reduce the diameter of the aorta to avoid stent un-anchoring.

3. Thank you for suggestion on issue of foreign body reaction and inflammation. Based on the subcutaneous implantation experiment of SHE biocompatibility evaluation in this study, the foreign body reaction induced by SHE was weak (Supplementary Fig. 8). But it is still worthwhile to further illustrate the inflammation of aneurysm when SHEs were applied in aneurysm wrapping. Hence, we used immunofluorescence labeling of CD3 and F4/80 to compare infiltration of lymphocytes and macrophages in each group.

According to the published studies²⁵⁻²⁸, the labeling of CD3 and F4/80 not only illustrate the degree of foreign body reaction but also reveal the effect of SHEs on aneurysm inflammation. The results show that the infiltration of lymphocytes and macrophages in SHE1 and SHE0.5 group were lower than aneurysm group, which illustrate the inflammations of aneurysm were alleviated in SHEs groups (Page13, line7, Supplementary Fig.13). Moreover, the apoptosis of vascular smooth muscle cell (VSMC) also contributes to the progress of aneurysm¹⁰. Hence, we future performed dual immunofluorescence labeling of α -SMA and caspase-3 and qRT-PCR of caspase-3 to observe the difference of VSMC apoptosis in each group. The results show that the counting number of apoptotic vascular smooth muscle cell in SHE1 and SHE0.5group were reduced when compared to aneurysm group, and the foldchange of caspase-3 transcription in SHE1 and SHE0.5 group were lower than that of aneurysm group (Page13, line9, Supplementary Fig. 14B).

In a summary, in addition to providing external mechanical support for the aneurysm, SHEs can also prevent the further expansion of the aneurysm by alleviating the inflammation of the artery wall and reducing the apoptosis of smooth muscle cells.

Supplementary Figure 13. Inflammation, apoptosis of vascular smooth muscle cell and neovascularization of aneurysm after SHE wrapping.

(A and B) Transverse sections of aorta were stained with CD3 and F4/80 to illustrate lymphocyte and macrophage infiltration in each group. Green marked lymphocyte and red marked macrophage. Scale bar = 40µm. (C) Dual labeling of α -SMA and caspase-3 in each group. Green marked vascular smooth muscle cell and red marked caspase-3. Scale bar = 40µm. (D) CD31 labeling to illustrate neovascular. The neovascular is marked with white arrow. Scale bar = 40 µm. (E through H) The statistic histogram of counting of lymphocyte, macrophage, apoptotic VSMC and neovascular respectively per high power filed (n = 5 in each group). *P < 0.05, ** P < 0.01, *** P < 0.001 **** P < 0.0001, ns = no significance.

mRNA transcriptional changes of caspase-3

GAPDH was used as internal reference. n = 5 in each group.

* p < 0.05, *** p < 0.001, **** p < 0.0001, ns = no significance.

7. Angiogenic processes have been shown to enhance inflammation and accelerate AAA in similar experimental models as well as benefit demonstrated with anti-angiogenic agents. What effects does SHE have on medial and peri-adventitial neovascularization?

Response:

Thank you for suggestion on issue of neovascularization. According to the previous researches²⁹⁻³¹, to better describe the effect of SHEs on neovascularization of aneurysm wall, we further performed the immunofluorescence labeling of CD31 and counted the number of neovessel in each group. The results show that the neovascularization was lower in group of SHE1, SHE0.5 than aneurysm group, while neovascularization of SHE 1 group was higher than that of SHE0.5 group (Page13, line9; Supplementary Fig.13D and 13H), which means that SHE wrapping can attenuate the angiogenic processes of aneurysm.

1. Bartoli, M.A., *et al.* In vivo assessment of murine elastase-induced abdominal aortic aneurysm with high resolution magnetic resonance imaging. *European journal of vascular and endovascular surgery : the official journal of the European Society for Vascular Surgery* **44**, 475-481 (2012).
2. Kusters, P.J.H., *et al.* CD40L Deficiency Protects Against Aneurysm Formation. *Arteriosclerosis, thrombosis, and vascular biology* **38**, 1076-1085 (2018).
3. Umebayashi, R., *et al.* Cilostazol Attenuates Angiotensin II-Induced Abdominal Aortic Aneurysms but Not Atherosclerosis in Apolipoprotein E-Deficient Mice. *Arteriosclerosis, thrombosis, and vascular biology* **38**, 903-912 (2018).
4. Martorell, S., *et al.* Vitamin D Receptor Activation Reduces Angiotensin-II-Induced Dissecting Abdominal Aortic Aneurysm in Apolipoprotein E-Knockout Mice. *Arteriosclerosis, thrombosis, and vascular biology* **36**, 1587-1597 (2016).
5. Lareyre, F., *et al.* TGF β (Transforming Growth Factor- β) Blockade Induces a Human-Like Disease in a Nondissecting Mouse Model of Abdominal Aortic Aneurysm. *Arteriosclerosis, thrombosis, and vascular biology* **37**, 2171-2181 (2017).

6. Krishna, S.M., *et al.* A peptide antagonist of thrombospondin-1 promotes abdominal aortic aneurysm progression in the angiotensin II-infused apolipoprotein-E-deficient mouse. *Arteriosclerosis, thrombosis, and vascular biology* **35**, 389-398 (2015).
7. Li, Z., *et al.* Runx2 (Runt-Related Transcription Factor 2)-Mediated Microcalcification Is a Novel Pathological Characteristic and Potential Mediator of Abdominal Aortic Aneurysm. *Arteriosclerosis, thrombosis, and vascular biology* **40**, 1352-1369 (2020).
8. Satta, J., *et al.* Chronic inflammation and elastin degradation in abdominal aortic aneurysm disease: an immunohistochemical and electron microscopic study. *European journal of vascular and endovascular surgery : the official journal of the European Society for Vascular Surgery* **15**, 313-319 (1998).
9. Sun, J., *et al.* Mast cells modulate the pathogenesis of elastase-induced abdominal aortic aneurysms in mice. *J Clin Invest* **117**, 3359-3368 (2007).
10. Golledge, J. Abdominal aortic aneurysm: update on pathogenesis and medical treatments. *Nat Rev Cardiol* **16**, 225-242 (2019).
11. Golledge, J., Krishna, S.M. & Wang, Y. Mouse models for abdominal aortic aneurysm. *Br J Pharmacol* (2020).
12. Anidjar, S., *et al.* Elastase-induced experimental aneurysms in rats. *Circulation* **82**, 973-981 (1990).
13. Bhamidipati, C.M., *et al.* Development of a novel murine model of aortic aneurysms using peri-adventitial elastase. *Surgery* **152**, 238-246 (2012).
14. Johnston, W.F., *et al.* Inhibition of Interleukin-1 Decreases Aneurysm Formation and Progression in a Novel Model of Thoracic Aortic Aneurysms. *Circulation* **130**, S51-S59 (2014).
15. Hawkins, R.B., *et al.* Mesenchymal Stem Cell Administration Attenuates Thoracic Aortic Aneurysm Formation. *Circulation* **134**, A16936-A16936 (2016).
16. Li, J., *et al.* IL (Interleukin)-33 Suppresses Abdominal Aortic Aneurysm by Enhancing Regulatory T-Cell Expansion and Activity. *Arteriosclerosis, thrombosis, and vascular biology* **39**, 446-458 (2019).
17. Lu, G., *et al.* A novel chronic advanced stage abdominal aortic aneurysm murine model. *J Vasc Surg* **66**, 232-242.e234 (2017).
18. Bi, Y., *et al.* Performance of a modified rabbit model of abdominal aortic aneurysm induced by topical application of porcine elastase: 5-month follow-up study. *European journal of vascular and endovascular surgery : the official journal of the European Society for Vascular Surgery* **45**, 145-152 (2013).
19. Busch, A., *et al.* Extra- and Intraluminal Elastase Induce Morphologically Distinct Abdominal Aortic Aneurysms in Mice and Thus Represent Specific Subtypes of Human Disease. *J Vasc Res* **53**, 49-57 (2016).
20. Gao, F., *et al.* Disruption of TGF- β signaling in smooth muscle cell prevents elastase-induced abdominal aortic aneurysm. *Biochemical and biophysical research communications* **454**, 137-143 (2014).
21. Daugherty, A., Manning, M.W. & Cassis, L.A. Angiotensin II promotes atherosclerotic lesions and aneurysms in apolipoprotein E-deficient mice. *J Clin Invest* **105**, 1605-1612 (2000).
22. Trachet, B., *et al.* Angiotensin II infusion into ApoE-/- mice: a model for aortic dissection

- rather than abdominal aortic aneurysm? *Cardiovasc Res* **113**, 1230-1242 (2017).
23. Wang, Y., Krishna, S. & Golledge, J. The calcium chloride-induced rodent model of abdominal aortic aneurysm. *Atherosclerosis* **226**, 29-39 (2013).
 24. Dhillon, J.S., Randhawa, G.K., Straehley, C.J. & McNamara, J.J. Late rupture after dacron wrapping of aortic aneurysms. *Circulation* **74**, 111-114 (1986).
 25. Annabi, N., *et al.* Engineering a highly elastic human protein-based sealant for surgical applications. *Sci Transl Med* **9**(2017).
 26. Raffort, J., *et al.* Monocytes and macrophages in abdominal aortic aneurysm. *Nature Reviews Cardiology* **14**, 457 (2017).
 27. Maegdefessel, L., *et al.* miR-24 limits aortic vascular inflammation and murine abdominal aneurysm development. *Nat Commun* **5**, 5214 (2014).
 28. Li, F.D., *et al.* Ablation and Inhibition of the Immunoproteasome Catalytic Subunit LMP7 Attenuate Experimental Abdominal Aortic Aneurysm Formation in Mice. *J Immunol* **202**, 1176-1185 (2019).
 29. Choke, E., *et al.* Abdominal aortic aneurysm rupture is associated with increased medial neovascularization and overexpression of proangiogenic cytokines. *Arteriosclerosis, thrombosis, and vascular biology* **26**, 2077-2082 (2006).
 30. Kaneko, H., *et al.* Resveratrol prevents the development of abdominal aortic aneurysm through attenuation of inflammation, oxidative stress, and neovascularization. *Atherosclerosis* **217**, 350-357 (2011).
 31. Yu, H., *et al.* Angiotensin II attenuates angiotensin II-induced aortic aneurysm and atherosclerosis in apolipoprotein E-deficient mice. *Sci Rep* **6**, 35190 (2016).

REVIEWERS' COMMENTS

Reviewer #3 (Remarks to the Author):

I have no further comments beyond my previous submission.

Reviewer #6 (Remarks to the Author):

The authors adequately addressed the comments of the reviewers.

Reviewer #7 (Remarks to the Author):

The authors have adequately responded to this reviewer's queries and the manuscript is much improved. Some of the responses in the Response to Reviewers are not adequately represented in the actual manuscript. In particular, the methodological details are important for the interpretation of presented data and should be included in the Supplement.

Dear Editors and Reviewers:

Thank you for your letter and for the reviewers' comments concerning our manuscript "The Self-healing Polyurethane-elastomer with Mechanical Tunability for Multiple Biomedical Applications In Vivo" (ID: NCOMMS-20-23626B). We have carefully studied these comments which are helpful for revising and improving our research.

The responses for comments of reviewers were listed orderly in following part of this letter. The content alterations in manuscript were marked with color of **RED**.

Reviewer #3 (Remarks to the Author):

I have no further comments beyond my previous submission.

Response:

Thank you for your work.

Reviewer #6 (Remarks to the Author):

The authors adequately addressed the comments of the reviewers.

Response:

Thank you for your work.

Reviewer #7 (Remarks to the Author):

The authors have adequately responded to this reviewer's queries and the manuscript is much improved. Some of the responses in the Response to Reviewers are not adequately represented in the actual manuscript. In particular, the methodological details are important for the interpretation of presented data and should be included in the Supplement.

Response:

Thank you for your suggestions. The main content of the responses in the *Response to Reviewers* have been described in the revised manuscript. The methodological details including MRI measurements (summarized in supplementary methods), morphology measures (summarized in supplementary methods), eNOS staining quantify (summarized in manuscript page 19, line30) and elastin grading method (summarized in supplementary methods) have been added in manuscript and supplementary information file methods part.